# Dynamic and timing properties of new aerosol particle formation and consecutive growth events

Imre Salma and Zoltán Németh

Institute of Chemistry, Eötvös University, H-1518 Budapest, P.O. Box 32, Hungary

Correspondence to: Imre Salma (salma@chem.elte.hu)

**Abstract.** Dynamic properties, i.e. particle formation rate $J_6$ and particle diameter growth rate $GR_{10}$, and timing properties, i.e. starting time ($t_1$) and duration time interval ($\Delta t$) of 247 quantifiable atmospheric NPF and growth events identified in the city centre and near-city background of Budapest over 6 full measurement years together with related gas-phase $H_2SO_4$ proxy, condensation sink (CS) of vapours, basic meteorological data and concentrations of criteria pollutant gases were derived, evaluated, discussed and interpreted. In the city centre, nucleation ordinarily starts at 09:15 UTC+1, and it is maintained for approximately 3 h. The NPF and growth events produce 4.6 aerosol particles with a diameter of 6 nm in 1 cm$^3$ of air in 1 s, and cause the particles with a diameter of 10 nm to grow with a typical rate of 7.3 nm h$^{-1}$. Nucleation starts approximately 1 h earlier in the near-city background, it shows substantially smaller $J_6$ (with a median of 2.0 cm$^{-3}$ s$^{-1}$) and $GR_{10}$ values (with a median of 5.0 nm h$^{-1}$), while the duration of nucleation is similar to that in the centre. Monthly distributions of the dynamic properties and daily maximum $H_2SO_4$ proxy do not follow the mean monthly pattern of the event occurrence frequency. The factors that control the event occurrence and that govern the intensity of particle formation and growth are not directly linked. New particle formation and growth processes advance in a different manner in the city and its close environment. This could likely be related to diversities in atmospheric composition, chemistry and physics. Monthly distributions and relationships among the properties mentioned provided indirect evidence that chemical species other than $H_2SO_4$ largely influence the particle growth and possibly atmospheric NPF process as well. The $J_6$, $GR_{10}$ and $\Delta t$ can be described by log-normal distribution function. Most extreme dynamic properties could not be explained by available single or compound variables. Approximately 40% of the NPF and growth events exhibited broad beginning, which can be an urban feature. For doublets, the later onset frequently shows more intensive particle formation and growth than the first onset by a typical factor of approximately 1.5. The first event is attributed to regional type, while the second event, superimposed on the first, is often associated with sub-regional, thus urban NPF and growth process.

## 1 Introduction

Molecules and molecular fragments in the air collide randomly and can form electrically neutral or charged clusters. Most clusters decompose shortly. Chemical stabilising interactions among certain components within a cluster can enhance its lifetime, during which it can grow further by additional molecular collisions through some distinguishable size regimes (Kulmala et al., 2014). If the diameter of these clusters reaches a critical value of 1.5±0.3 nm (Kulmala et al., 2013), they become thermodynamically stable, and their further growth turns into a spontaneous process. Supersaturation is a necessary atmospheric condition for this principal transformation. It is essentially a phase transition, which takes place in a dispersed manner in the atmosphere, so it generates an aerosol system. The newly formed particles grow further by condensation to larger sizes in most cases due to the existing supersaturation. Photochemical oxidation products such as $H_2SO_4$ (Sipilä et al., 2010), extremely low-volatile organic compounds (ELVOCs, Ehn et al., 2014; Jokinen et al., 2015) and highly oxygenated molecules (HOMs, Bianchi et al., 2016; Kirkby et al., 2016; Tröstl et al., 2016) together with $H_2O$ vapour, $NH_3$ (Kirkby et al., 2011), amines (Almeida et al., 2013), other oxidation products of volatile organic compounds (VOCs; Metzger et al., 2010; Schobesberger et al., 2013; Riccobono et al., 2014) and some inhibiting chemical species (e.g. isoprene or $NO_2$; Kiendler-Scharr et al., 2009; Kerminen et al., 2018) can play an important role in particle formation and growth. The VOCs include compounds of both anthropogenic and biogenic origin, mainly isoprenoids such as α-pinene (Kirkby et al., 2016). In some specific coastal regions, iodine oxides produced from marine biota are involved (O'Dowd et al., 2002). Atmospheric concentration of these key compounds at a level that is smaller by 12–14 orders of magnitude than the concentration of air molecules is already sufficient for the phenomenon (Kulmala et al., 2014). Relative importance of the organics increases with particle size (Riipinen et al., 2011; Ehn et al., 2014), and their supersaturation is maintained by fast gas-phase autooxidation reactions of VOCs (Crounse et al., 2013). The overall phenomenon is ordinarily confined in time for 1 day or so, and, therefore, it can be regarded as an event in time, and is referred as new aerosol particle formation (NPF) and consecutive particle diameter growth event.


Such events appear to take place almost everywhere in the world and anytime (Kulmala et al., 2004; Kerminen et al., 2018; Nieminen et al., 2018). Their occurrence frequency and, more importantly, their contribution to particle number concentrations were found to be substantial

or determinant in the global troposphere (Spracklen et al., 2006; Kulmala et al., 2014).
Moreover, their contribution to the number of cloud condensation nuclei (CCN) can be 50% or
even more (Makkonen et al., 2009; Merikanto et al, 2009; Sihto et al., 2011), which links the
events to climate system, and emphasizes their global relevance (Kerminen et al., 2012;
Makkonen et al., 2012; Carslaw et al., 2013; Gordon et al., 2016). New particle formation and
growth events were proved to be common in polluted air of large cities as well with a typical
relative occurrence frequency between 10% and 30% (Woo et al., 2001; Baltensperger et al.,
2002; Alam, et al., 2003; Wehner et al., 2004; Salma et al., 2011; Dall'Osto et al., 2013; Xiao
et al., 2015; Zhang et al., 2015; Kulmala et al., 2017, Nieminen et al., 2018). The coupling and
relationships between regional and urban (sub-regional) NPF were demonstrated at least under
favourable orographic conditions (Salma et al., 2016b). New particle formation can increase
the existing particle number concentrations in city centres by a factor of approximately 2 on
nucleation days, while it can produce 13–28% of ultrafine (UF) particles as a lower estimate on
a longer (e.g. annual) time scale (Salma et al., 2017). Particle concentrations from NPF are also
important when compared to (primary) particles emitted by their dominant source in cities,
namely by road vehicles with internal combustion engines (Paasonen et al., 2016). These results
jointly suggest that particles from NPF and growth events in cities can influence not only the
urban climate but can contribute to the public's excess health risk from particle number
exposures (Oberdörster et al., 2005; Braakhuis et al., 2014; Salma et al., 2015), and,
furthermore, could be linked to the role of human actions in all these effects.

Despite these potentials, conclusive interpretation of the data obtained, and results derived
specifically for cities remained hindered so far. Several-year long, semi-continuous, critically
evaluated, complex and coherent data sets are required for this purpose, which have been
generating gradually. As part of this international progress, investigations dedicated to urban
NPF and growth events in Budapest have been going on since November 2008. Measurements
for 5 full years were realised in the city centre at a fixed location, 1 full year was devoted to
measurements in a near-city background environment, and some other measurements were
accomplished in different urban microenvironments for time intervals of a few months. The
main objectives of this study are to determine, present and analyse the dynamic properties, i.e.
particle formation rate and particle diameter growth rate, timing properties, i.e. starting time
and duration time interval of nucleation process of NPF and growth events together with the
major sources and sink of condensing vapours, basic meteorological data and criteria pollutant
gases for 6 years, to investigate and interpret their relationships, to discuss their monthly
distributions, to evaluate and detect some of their features specific for urban atmospheric
environments, and to demonstrate some specific urban influence on the calculation of the
properties. These quantities and relationships are of basic importance in many atmospheric
processes for several reasons. Our goals are in line with the research needs for global
atmospheric nucleation studies (Kerminen et al., 2018; Nieminen et al., 2018).

**2 Experimental methods**

The measurements took place at two urban locations in Budapest, Hungary. Most measurements
were realised at the Budapest platform for Aerosol Research and Training (BpART) facility (N
47° 28' 29.9", E 19° 3' 44.6", 115 m above mean sea level (a.s.l.; Salma et al., 2016a). This site
represents a well-mixed, average atmospheric environment for the city centre. The other
location was situated at the NW border of Budapest in a wooded area of the Konkoly
Astronomical Observatory of the Hungarian Academy of Sciences (N 47° 30' 00.0", E 18° 57'
46.8", 478 m a.s.l.). This site characterises the air masses entering the city since the prevailing
wind direction in the area is NW. The measurements were accomplished for 6 full-year long
time intervals, i.e. from 03–11–2008 to 02–11–2009, from 19–01–2012 to 18–01–2013, from
13–11–2013 to 12–11–2014, from 13–11–2014 to 12–11–2015, from 13–11–2015 to 12–11–
2016 and from 28–01–2017 to 27–01–2018. In the measurement year 2012–2013, the
instruments were set up in the near-city background, while in all other years, they were installed
in the city centre. Local time (LT=UTC+1 or daylight-saving time, UTC+2) was chosen as the
time base of the data unless otherwise indicated because it had been observed in earlier
investigations that the daily activity time pattern of inhabitants substantially influences many
atmospheric processes in cities (Salma et al., 2014; Sun et al., 2019).

The main measuring system was a flow-switching type differential mobility particle sizer
(DMPS). It consists of a radioactive ($^{60}$Ni) bipolar charger, a Nafion semi-permeable membrane
dryer, a 28-cm long Vienna-type differential mobility analyser and a butanol-based
condensation particle counter (TSI, model CPC3775). The sample flow was 2.0 L min$^{-1}$ in the
high-flow mode, and 0.31 L min$^{-1}$ in the low-flow mode with sheath air flow rates 10 times
larger than for the sample flows. The DMPS measures particle number concentrations in an
electrical mobility diameter range from 6 to 1000 nm in the dry state of particles (with a relative
humidity of RH<30%) in 30 channels, which finally yields 27 channels after averaging 3
overlapping channels when joining the data for the 2 flow modes. The time resolution of the

measurements was approximately 10 min till 18–01–2013, and 8 min from 13–11–2013 (after a planned update of the DMPS system). There was no upper size cut-off inlet applied to the sampling line, and a weather shield and insect net were only attached. The sampling inlets were identical at both locations except for the height of the installation above the ground, which was 12.5 m in the city centre and approximately 1.7 m in the near-city background. The measurements were performed according to the international technical standard (Wiedensohler et al., 2012). The availability of the DMPS data over 1-year long time intervals are summarised in Table 1.

Synoptic meteorological data for air temperature ($T$), RH, wind speed (WS) and wind direction (WD) were obtained from a measurement station of the Hungarian Meteorological Service (HMS, station no. 12843) by standardised methods with a time resolution of 1 h. Global solar radiation (GRad) data were measured by the HMS at a distance of 10 km in E direction with a time resolution of 1 h. Meteorological data were available in >90% of the possible cases in each year. Concentrations of $SO_2$, $O_3$, $NO_x$ and CO were obtained from measurement stations of the National Air Quality Network in Budapest (in a distance of 4.5 km from the urban site, and of 6.9 km from the near-city background site) located in the upwind prevailing direction from the measurement sites. They are measured by UV fluorescence (Ysselbach 43C), UV absorption (Ysselbach 49C), chemiluminescence (Thermo 42C) and IR absorption methods (Thermo 48i), respectively with a time resolution of 1 h. The concentration data were available in >85% of the yearly time intervals, and >98% of them were above the limit of determinations (LOD). It is worth mentioning that the LOD of the $SO_2$ analyser was approximately 0.2 µg m$^{-3}$, and that the hourly average $SO_2$ concentration in the Budapest area is ordinarily distributed without larger spatial gradients (Salma et al., 2011). For the present study, this was proved by evaluating the concentration ratios from 2 different municipal stations which are in the closest distance from the BpART facility in 2 different directions with an angle of 60° between them. The mean $SO_2$ concentration ratio and standard deviation (SD) for the 2 stations were 81±20% over the 5-year long measurement time interval.

**3 Data treatment**

The measured DMPS data were evaluated according to the procedure protocol recommended by Kulmala et al. (2012) with some refinements that are related to urban features (see Sect. 3.1). Particle number concentrations in the diameter ranges from 6 to 1000 nm ($N$), from 6 to 25 nm

($N_{6-25}$), from 6 to 100 nm ($N_{6-100}$ or UF particles) and from 100 to 1000 nm ($N_{100-1000}$) were
calculated from the measured and inverted DMPS concentrations. Particle number size
distribution surface plots showing jointly the variation in particle diameter and particle number
concentration density in time were also derived. Identification and classification of NPF and
growth events was accomplished on these surface plots (Dal Maso et al., 2005; Németh et al.,
2018) on a daily basis into the following main classes: NPF event days, non-event days, days
with undefined character, and days with missing data (for more than 4 h during the midday).
Relative occurrence frequency of events was determined for each month and year as the ratio
of the number of event days to the total number of relevant (i.e. all−missing) days. A subset of
NPF events with uninterrupted evolution in time, which are called quantifiable (class 1A)
events, were further separated because the time evolution of their size distribution functions
was utilised to determine the dynamic and timing properties with good accuracy and reliability.

**3.1 Dynamic and timing properties**

Growth rate (GR) of nucleation-mode particles was calculated by mode-fitting method
(Kulmala et al., 2012). Particle number median mobility diameter (NMMD) of the nucleation
mode were obtained from fitting the individual size distributions by DoFit algorithm (Hussein
et al., 2004). The growth rate was determined as the slope of the linear line fitted to the time
series of the NMMD data within a time interval around a diameter $d$, where the dependency
could be satisfactorily approximated by linear fit. Since the nucleation mode was mostly
estimated by $N_{6-25}$ in the calculations of the formation rate (see below), and since the median
of the related diameter interval (from 6 to 25 nm) is close to $d=10$ nm, GRs for particles with a
diameter of 10 nm were determined ($GR_{10}$). This type of GR can be interpreted as an average
GR as far as the given particle diameter range is concerned, but it actually expresses the
beginning of the growth process only. Particle growth can slow substantially in time in specific
cases, and this can affect considerably the formation rate calculations (see later).

Time evolution of an aerosol population is described by the general dynamic equation which
was rearranged, simplified and approximated by several quantities (Kulmala et al., 2001; Dal
Maso et al., 2002; Kulmala et al., 2012; Cai and Jiang, 2017) to express the formation rate $J_6$
of particles with the smallest detected diameter of $d_{min}=6$ nm in a form utilised in the present
evaluation as:

$$J_6 = \frac{\mathrm{d}N_{6\text{-}25}}{\mathrm{d}t} - \frac{\mathrm{d}N_{\mathrm{Ai},<25}}{\mathrm{d}t} + \mathrm{CoagS}_{10}(N_{6\text{-}25} - N_{\mathrm{Ai},<25}) + \frac{\mathrm{GR}_{10}}{(25-6)}(N_{6\text{-}25} - N_{\mathrm{Ai},<25})\,. \tag{1}$$

The first term on the right side of Eq. 1 expresses the concentration increment. The particle
number concentration in the size range from 6 to 25 nm (i.e. $N_{6-25}$) is usually selected to
approximate the nucleation-mode particles $N_{\mathrm{nuc}} \approx N_{6-25}$. This is a reasonable choice because it
was proved to be advantageous and effective way in handling fluctuating data sets since $N_{6-25}$
often exhibits smaller scatter in time and less sensitivity than the fitted area of the nucleation
mode. It is implicitly assumed that the intensity of the NPF is constant for a certain time interval,
and, therefore, $\mathrm{d}N_{6-25}/\mathrm{d}t$ can be determined as the slope of the linear function of $N_{6-25}$ versus
time $t$ within an interval where the dependence could be satisfactorily approximated by linear
fit. A limitation of the relatively wide size range (6–25 nm) selected can be manifested by
disturbances from primary particles particularly in urban environments. This is taken into
account by an additional term of $N_{\mathrm{Ai},<25}$, which is discussed below.

The second term on the right side of Eq. 1 expresses the contribution of high-temperature
emission sources, usually of vehicular road traffic (Paasonen et al., 2016; Salma et al., 2017) to
$N_{6-25}$, which can provisionally disturb the assumption of $N_{\mathrm{nuc}} \approx N_{6-25}$. A typical example of such
a situation is shown in Fig. S1a from 10:09 to 12:23 LT. In these specific cases, the contribution
of primary emissions was estimated from the slope of the time series of the fitted peak area of
the Aitken mode below $d<25$ nm ($N_{\mathrm{Ai},<25}$) in the time region under consideration. Reliable
separation of the nucleation and Aitken modes from each other was hindered or was not possible
for a few individual size distributions due to overlapping modes and the scatter in the measured
concentration data, and these individual cases were excluded from or skipped in the time series.

The third term on the right side of Eq. 1 represents the loss of particles due to coagulation
scavenging (with pre-existing particles). The coagulation scavenging efficiency for particles
with a diameter of 10 nm ($\mathrm{CoagS}_{10}$) was selected to approximate the mean coagulation
efficiency of nucleation-mode particles ($\mathrm{CoagS}_{\mathrm{nuc}}$). This diameter was chosen by considering
the median of the related diameter range, which was discussed above for GR. The coagulation
efficiency was calculated from classical aerosol mechanics with adopting a mass
accommodation coefficient of 1 and utilizing the Fuchs' transition-regime correction factor
(Kulmala et al., 2001; Dal Maso et al., 2005; Kulmala et al., 2013) by using computation scripts
developed at the University of Helsinki. Self-coagulation within the nucleation mode was
neglected due to limited ambient concentrations. Hygroscopic growth of particles was not
considered since this depends on chemical composition of particles, which is unknown.

The fourth term on the right side of Eq. 1 expresses the growth out of newly formed particles
from the size range by condensation of vapours. The $GR_{10}$ was selected to approximate a
representative value at the median of the particle diameter range considered (Vuollekoski et al.,
2012). It is implicitly assumed that $GR_{10}$ can be regarded to be constant over the time interval
under consideration. Nevertheless, the growth of nucleation-mode particles in time is
occasionally limited (Fig. S1b). In these specific cases, the mean relative area of the nucleation
mode below 25 nm was determined by fitting individual size distributions around the time of
the maximum nucleation-mode NMMD, and the ratios were averaged. A correction in form of
the mean relative area was adopted as a multiplication factor for the growth out term in Eq. 1.
On very few days, the growth of newly formed particles was followed by a decrease in
nucleation-mode NMMD (Salma et al., 2016a). In these cases, the shrinkage rate (with a formal
$GR_{10} < 0$) was derived and adopted in Eq. 1. Relative contributions of the concentration
increment coagulation loss and growth out from the diameter interval to $J_6$ are decreasing in
this order with mean values of 71%, 17% and 12%, respectively (Table S1).

The formation and growth rates for the measurement years of 2008–2009 and 2012–2013 were
calculated earlier by a slightly different way and neglecting the urban features discussed above
(Salma et al., 2011, 2016b). To obtain consistent data sets, the dynamic properties for these 2
years were re-evaluated by adopting the present improved protocol and implementing the
experience gained over the years. The mean new-to-old rate ratios with SDs for the $GR_{10}$ and
$J_6$ were 1.06±0.32 and 1.23±0.37, respectively in the city centre (2008–2009) and 1.04±0.21
and 1.20±0.35, respectively in the near-city background (2012–2013). It was the smaller rates
that were primarily and sometimes substantially impacted. The modifications were
simultaneously adopted. The subtraction of particle number concentrations emitted by road
traffic from $N_{6-25}$ usually leads to a decrease in the coagulation loss term and loss term due to
growth out from the diameter range of 6–25 nm. At the same time, the subtraction can also
influence the slope of the concentration change in time ($dN_{nuc}/dt$) depending on the actual time
evolution of perturbing emission source. In addition to that, the time interval in which this slope
is considered to be constant was set in a new treatment. It is noted that the relative contributions
of the concentration increment, coagulation loss and growth out from the diameter interval to
$J_6$ have different weights in propagating their effects. Furthermore, $J_6$ itself also depends on
GR$_{10}$, which makes the relationships even more complex. These connected effects explain why
the changes resulted in increments. The re-calculation is considered as a methodological
improvement over the years of research.

The assumptions and estimations above usually represent a reasonable approximation to reality.
The $N_{6–25}$ is derived from the experimental data in a straightforward way, the GR$_{10}$ and the
corrections for primary particles and limited particle growth depend on the quality of the size
distribution fitting as well, while the CoagS$_{10}$ is determined by using a theoretical model. The
resulting accuracies of the dynamic properties, in particular of $J_6$, look rather complicated. They
also depend on the spatial heterogeneity in the investigated air masses particularly for the
observations performed at the fixed site, size and time resolution of the concentrations
measured, diameter range of the size distributions, fluctuations in the experimental data,
selection of the particle diameter interval, choice of the time interval of interest (for linear fits),
sensitivity of the models to the input uncertainties (Vuollekoski et al., 2012), and also on the
extent of the validity of the assumptions applied under highly polluted conditions (Cai and
Jiang, 2017). The situation is further complicated with the fact that the dynamic (and also the
timing) properties are connected to each other. Finally, it is important to recognise that some
NPF and growth curves on the surface plots have rather broad starting time interval (Fig. S1b
and S1c). They occur in a considerable abundance in cities, e.g. in 40% of all quantifiable events
in Budapest (Sect. 4.4). This may yield badly defined or composite dynamic properties, whose
uncertainty can have principle limitations which can prevail on the experimental and model
uncertainties.

Timing properties of NPF and growth events are increasingly recognised, and they can provide
valuable information even if they are estimated indirectly from the observed diameter interval
>1.5 nm (Sect. 1). The earliest estimated time of the beginning of a nucleation ($t_1$) and the latest
estimated time of the beginning of a nucleation ($t_2$) were derived by a comparative method
(Németh and Salma, 2014) based on the variation in the content of the first size channel of the
DMPS system. Both time parameters include a time shift that accounts for the particle growth
from the stable neutral cluster mode at approximately 2 nm to the smallest detectable diameter
limit of the DMPS systems (6 nm in our case) by adopting the GR value in the size window
nearest to it in size space. The difference $\Delta t = t_2 – t_1$ is considered as the duration time interval of
the nucleation process. It represents the time interval during which new aerosol particles are
generated in the air. The timing properties are expressed in UTC+1, and their uncertainty is
regarded to be ca. 30 min under ordinary NPF and growth situations.

**3.2 Sources and sink**

Relative effects and role of gas-phase $H_2SO_4$ were estimated by its proximity measure (proxy
value) containing both its major source and sink terms under steady-state conditions according
to Petäjä et al. (2009). It was calculated for GRad>10 W m$^{-2}$. Formaly, it is possible to convert
the $H_2SO_4$ proxy values to $H_2SO_4$ concentrations by an empirical scaling factor of $k$=1.4×10$^{-7}$×GRad$^{-0.70}$, where GRad is expressed in a unit of W m$^{-2}$ (Petäjä et al., 2009). The factor was,

however, derived for a remote boreal site, and, therefore, we prefer not to perform the
conversion since urban areas are expected to differ from the boreal regions. The conversion was
applied only to estimate the order of average $H_2SO_4$ atmospheric concentration levels. The
results derived by utilising the proxy are subject to larger uncertainties than for the other
properties because of these limitations, but they may indicate well gross tendencies.

Condensation sink for vapour molecules onto the surface of existing aerosol particles was
computed for discrete size distributions as described in earlier papers (Kulmala et al., 2001; Dal
Maso et al., 2002, 2005) and summarised by Kulmala et al. (2013). The equilibrium vapour
pressure of the condensing species was assumed to be negligible at the surface of the particles,
thus similar to sulfuric acid. Dry particle diameters were considered in the calculations.

**4 Results and discussion**

Annual median total particle number concentrations ($N$) for each measurement year are
summarised in Table 1. The data for the city centre indicate a moderate decreasing trend. The
mean UF/$N$ ratio with SD for the same measurement time intervals were 67±14%, and 79±6%,
75±10%, 75±11%, 76±11% and 80±10%, respectively. The values correspond to ordinary
urban atmospheric environments in Europe (Putaud et al., 2010, Sun et al., 2019). An overview
on the number of classified days for each measurement year is also given in Table 1. The
availability of the daily size distribution surface plots with respect to all days ensures that the
data are representative on yearly and monthly time scales, except for the months August and
September 2015, when there were missing days in larger ratios. The number of quantifiable
event days (248 cases) is also considerable, which establishes to arrive at firm conclusion for
the NPF and growth events as well.
**Table 1.** Annual median total particle number concentrations (in $10^3$ cm$^{-3}$), number of days with NPF
and growth event, quantifiable event days, non-event days, undefined days, missing days and the
coverage (in %) of relevant days in the near-city background and city centre separately for the 1-year
long measurement time intervals.

| Environment | Background | Centre | | | | |
|---|---|---|---|---|---|---|
| Time interval | 2012–2013 | 2008–2009 | 2013–2014 | 2014–2015 | 2015–2016 | 2017–2018 |
| Concentration | 3.4 | 11.5 | 9.7 | 9.3 | 7.5 | 8.7 |
| Event days | 96 | 83 | 72 | 81 | 35 | 83 |
| Quantifiable days | 43 | 31 | 48 | 56 | 18 | 52 |
| Undefined days | 19 | 34 | 24 | 25 | 8 | 23 |
| Non-event days | 231 | 229 | 267 | 240 | 226 | 257 |
| Coverage | 95 | 95 | 99 | 95 | 73 | 99 |
| Missing days | 20 | 19 | 2 | 19 | 97 | 2 |

It was previously shown that the NPF and growth events observed in the city centre of Budapest
and its background ordinarily happen above a larger territory or region in the Carpathian Basin
(Németh and Salma, 2014), and they are linked to each other as a spatially coherent and joint
atmospheric phenomenon (Salma et al., 2016b). From the point of the occurrence frequency
distribution, they can, therefore, be evaluated jointly in the first approximation. An overall
monthly mean relative occurrence frequency of nucleation days derived for all 6 measurement
years is shown in Fig. 1. The annual mean frequency with SD was 22±5%, which is considerable
and is in line with other urban sites (Sect. 1). The monthly mean frequency has a temporal
variation, which can be characterised by a noteworthy pattern. The mean monthly dependency
exhibits an absolute and a local minimum in January (5.6%) and August (21%), respectively,
and an absolute and a local maximum in April (40%), and September (31%), respectively.
Nevertheless, the SDs of the monthly means indicate prominent variability from year to year.
The pattern can be related to multivariate relationships and complex interplay among the
influencing factors, which include the air temperature (January is the coldest month, while
August is the warmest month in the Carpathian Basin) and enhanced emission of biogenic
VOCs in springtime (March–April) and early autumn (September) as well (Salma et al., 2016b).
It is noted that the findings derived for the separate city-centre data set are very similar to the
results presented above.

**Figure 1.** Monthly mean relative occurrence frequency of NPF and growth events for the joint 6-year
long data set. The error bars show ±1 standard deviation, the horizontal line in cyan indicates the overall
annual mean frequency, the yellow bands represent ±1 standard deviation of the annual mean, and the
smooth curve in red serves to guide the eye.

The properties and variables studied were derived in full time resolution. They were averaged
in several ways for different conditions and for various purposes to obtain typical average
descriptive characteristics. In 1 case (31–08–2016), the NPF and growth event could reliable
be identified, while the measured absolute particle number concentrations could not be
validated due to experimental troubles, and, therefore, it was left out from the further
calculations. Similarly, there were 1 and 4 events with unusually/extraordinarily large dynamic
properties in the measurement years 2014–2015 and 2017–2018, respectively. More
specifically, 5 individual $J_6$ data when expressed in a unit of $cm^{-3}$ $s^{-1}$ and 1 individual $GR_{10}$
data when given in nm $h^{-1}$ were >20 (Table 3). These extremes were left out from the overview
statistics to maintain the representativity (they could be influenced by some unknown extra or
very local sources) and to fulfil better the basic requirements of correlation analysis. If an event
showed a double beginning then the dynamic properties for the first onset were considered in
the basic overview since this onset is of regional relevance (Salma et al., 2016b). The extreme
NPF and growth events and the characteristics for the second onsets were, however, evaluated
separately and are discussed in detail and interpreted in Sect. 4.4.

**4.1 Ranges and averages**

Ranges and averages with SDs of formation rate $J_6$, growth rate $GR_{10}$, starting time of
nucleation ($t_1$) and duration time interval of nucleation ($\Delta t$) are summarised in Table 2 for
separate measurement years and for the joint 5-year long city centre data set. In the city centre,
nucleation generally starts at 09:15 UTC+1, and it is typically maintained for approximately 3
h. The NPF and growth events ordinarily produce 5.6 new aerosol particles with a diameter of
6 nm in 1 $cm^3$ of air in 1 s, and cause the particles with a diameter of 10 nm to grow with a
typical rate of 7.6 nm $h^{-1}$. The statistics for $J_6$ and $GR_{10}$ are based on 199 and 203 events,
respectively. The corresponding data for the separate years show considerable variability
without obvious trends or tendencies. The differences between the years can likely be related
to changes in actual atmospheric chemical and physical situations and conditions, and to the
resulting modifications in the sensitive balance and delicate coupling among them from year to
year. Spread of the individual data for $GR_{10}$ is smaller than for $J_6$; the relative SDs for the joint
5-year long city centre data set were 38% and 68%, respectively.

The dynamic properties and $t_1$ data tend to be smaller in the near-city background than in the
city centre. In general, nucleation starts 1 h earlier in the background, and the events typically
show significantly smaller $J_6$ (with a median of 2.0 cm$^{-3}$ s$^{-1}$) and $GR_{10}$ (with a median of 5.0
nm h$^{-1}$). Duration of the nucleation is very similar to that in the city centre. All starting times
of nucleation were larger than (in a few cases, very close to) the time of the sunrise. This implies
that no nocturnal NPF and growth event has been identified in Budapest so far. The particle
growth process (the so-called banana curve) could be traced usually for a longer time interval
(up to 1.5 d) in the background than in the centre.

These results are in line with the ideas on atmospheric nucleation and consecutive particle
growth process (e.g. Kulmala et al., 2014; Zhang et al., 2015; Kerminen et al., 2018). It was
observed in a recent overview study (Nieminen et al., 2018) that the formation rate of 10–25
nm particles increased with the extent of anthropogenic influence, and in general, it was 1–2
orders of magnitude larger in cities than at sites in remote and clean environments.

**Table 2.** Ranges, averages and standard deviations of aerosol particle formation rate $J_6$, particle diameter
growth rate $GR_{10}$, starting time ($t_1$) and duration time interval ($\Delta t = t_2 - t_1$) of nucleation process of
quantifiable NPF and growth events in the near-city background and city centre separately for the 1-
year long measurement time intervals and for the joint 5-year long city centre data set.

| Environment | Background | Centre | | | | | |
|---|---|---|---|---|---|---|---|
| Time interval | 2012–2013 | 2008–2009 | 2013–2014 | 2014–2015 | 2015–2016 | 2017–2018 | All 5 years |
| Formation rate $J_6$ (cm$^{-3}$ s$^{-1}$) | | | | | | | |
| Minimum | 0.48 | 1.47 | 1.13 | 0.81 | 1.19 | 1.60 | 0.81 |
| Median | 2.0 | 4.2 | 3.5 | 4.4 | 4.6 | 6.3 | 4.6 |
| Maximum | 5.6 | 15.9 | 17.8 | 18.0 | 15.3 | 17.3 | 18.0 |
| Mean | 2.2 | 4.7 | 5.2 | 5.6 | 5.0 | 6.6 | 5.6 |
| St. deviation | 1.3 | 2.6 | 3.7 | 4.2 | 3.7 | 3.3 | 3.8 |
| Growth rate $GR_{10}$ (nm h$^{-1}$) | | | | | | | |
| Minimum | 3.0 | 3.7 | 3.1 | 2.8 | 3.2 | 3.3 | 2.8 |
| Median | 5.0 | 7.6 | 6.6 | 6.5 | 8.0 | 7.5 | 7.3 |
| Maximum | 9.8 | 17.4 | 19.0 | 18.0 | 15.5 | 19.8 | 19.8 |
| Mean | 5.2 | 7.8 | 7.2 | 7.3 | 7.7 | 8.0 | 7.6 |
| St. deviation | 1.4 | 2.6 | 2.8 | 3.2 | 3.0 | 2.8 | 2.9 |
| Starting time, $t_1$ (HH:mm UTC+1) | | | | | | | |
| Minimum | 05:51 | 07:14 | 06:44 | 05:48 | 07:31 | 05:57 | 05:48 |
| Median | 08:19 | 09:26 | 09:22 | 08:48 | 09:45 | 09:18 | 09:15 |
| Maximum | 11:09 | 11:38 | 12:21 | 11:23 | 12:45 | 12:15 | 12:45 |
| Mean | 08:17 | 09:27 | 09:25 | 08:49 | 10:02 | 09:24 | 09:19 |
| St. deviation | 01:11 | 01:05 | 01:26 | 01:22 | 01:23 | 01:36 | 01:26 |
| Duration time, $\Delta t$ (HH:mm) | | | | | | | |
| Minimum | 01:23 | 00:52 | 00:42 | 00:31 | 01:03 | 01:26 | 00:31 |
| Median | 03:16 | 02:36 | 02:04 | 03:53 | 02:31 | 03:49 | 02:57 |
| Maximum | 06:44 | 06:04 | 05:34 | 07:46 | 06:05 | 07:55 | 07:55 |
| Mean | 03:30 | 02:44 | 02:14 | 03:52 | 02:58 | 03:57 | 03:18 |
| St. deviation | 01:40 | 01:11 | 01:01 | 01:40 | 01:47 | 01:39 | 01:40 |


Ranges and averages with SDs of some related atmospheric properties, namely of mean CS
averaged for the time interval from $t_1$ to $t_2$, daily maximum gas-phase $H_2SO_4$ proxy, daily mean
$T$ and RH (Table S2), and of daily median concentrations of $SO_2$ (as the major precursor of gas-
phase $H_2SO_4$), $O_3$ (as an indicator of photochemical activity), $NO_x$ and CO gases (as indicators
of anthropogenic combustion activities and road vehicle emissions) (Table S3) were also
derived for quantifiable NPF and growth event days, and are further evaluated. The annual
mean CS values exhibited decreasing tendency in the city centre over the years. The individual
values remained below approximately $20 \times 10^{-3}$ s$^{-1}$, which agrees well with the results of our
earlier study (Salma et al., 2016b) according to which the CS suppresses NPF above this level
in the Carpathian Basin. Maximum $H_2SO_4$ proxy values reached substantially higher levels (by
a factor of approximately 2) in the near-city background than in the city centre due mainly to
the differences in the CS and [$SO_2$]. The differences between the 2 sites are particularly evident
when considering their smallest values. The largest variability in the annual average values
were observed for the proxy. Median concentration of $H_2SO_4$ molecules was roughly estimated
to be approximately $5 \times 10^5$ cm$^{-3}$ by adopting the scaling factor, although it is largely uncertain
due to the limitations of the factor (Sect. 3.2). The air $T$ displayed quite similar and comparable
values over the years at both sites. The discussion of its overall effect on the dynamic properties
is accomplished in Sec. 4.2, where the monthly distributions are presented. Some events
happened at daily mean temperatures below zero. The daily mean RH and its SD for the city
centre and near-city background were $54\pm11\%$ and $64\pm12\%$, respectively. There were events
that occurred at RHs as high as 90%. Relationships of the dynamic properties with $T$ and RH
are also obscured with strong seasonal cycle of these meteorological data and with the fact that
air masses arriving to the receptor site in different trajectories are often characterised by distinct
levels of meteorological data.

As far as the pollutant gases are concerned (Table S3), $SO_2$ showed somewhat smaller daily
median values, and $O_3$ exhibited substantially smaller levels on event days in the city centre
than in the near-city background, while concentrations of $NO_x$ and CO were obviously larger
in the city than in its close background. The differences can primarily be explained by intensity
and spatial distribution of their major sources and atmospheric chemical reactions, and the
joined concentration data resembles typical situations without photochemical smog episodes in
cities. There was no obvious decrease in $SO_2$ concentration during these years in contrast with
an earlier decreasing trend from mid-1980s till about 2000.

**4.2 Monthly distributions**

Distributions of the monthly mean $J_6$, $GR_{10}$, daily maximum gas-phase $H_2SO_4$ proxy, mean CS,
daily mean air $T$ and RH, and daily median $SO_2$, $O_3$, $NO_x$ and CO concentrations for quantifiable
NPF and growth events for the joint city centre data sets are shown in Fig. 2. The distributions
– eminently for $J_6$, $GR_{10}$, $H_2SO_4$ proxy and $SO_2$ – do not follow the monthly pattern of the event
occurrence frequency at all (cf. Fig. 1). Instead, the $J_6$, $GR_{10}$ and $H_2SO_4$ proxy tend to exhibit

larger values in summer months, and they temporal changes over the other months are smooth and do not show distinctive features. The elevations are substantial; the estimated maximum level was larger than the baseline by a factor of 2.1 for the $J_6$, and by a factor of approximately 1.4 for the $GR_{10}$ and $H_2SO_4$ proxy. Intensity of solar radiation, its seasonal cycling, concentration of atmospheric precursors in different months, biogenic processes, anthropogenic activities and the fact that rate coefficients of many thermal chemical/physicochemical processes in the nature (including GR, Paasonen et al., 2018) increase with $T$ could play an important role in explained the distributions.

**Figure 2.** Distribution of monthly mean aerosol particle formation rate $J_6$ in a unit of $cm^{-3} s^{-1}$ and particle diameter growth rate $GR_{10}$ in a unit of $nm h^{-1}$ (a), mean condensation sink for vapours (CS) in a unit of $s^{-1}$ averaged over the nucleation time interval ($t_1$, $t_2$) and daily maximum gas-phase $H_2SO_4$ proxy in a unit of $\mu g \, m^{-5}$ W s (b), daily mean air temperature ($T$) in a unit of °C and daily mean relative humidity (RH) in % (c), and daily median concentrations of $SO_2$, $O_3$, $NO_x$ and CO for quantifiable NPF and growth events in the city centre for the joint 5-year long time interval. The error bars are shown for one side and indicate 1 standard deviation. Number of the individual data averaged in each month is displayed next to the symbols. The horizontal lines indicate the overall mean. The nonlinear curves assist to guide the eye.

The differences in the GRad (and some other properties) are, however, biased by the seasonal cycle of solar electromagnetic radiation via the seasonal variation of NPF occurrence frequency. Nevertheless, the misalignment among the monthly distributions of NPF and growth event occurrence frequency and all the other properties indicates that the occurrence or its basic causes are not linked with the dynamic properties in a straightforward or linear manner in the Carpathian Basin including Budapest.

Some of our results are in line with other observations according to which GR exhibited almost exclusively a summer maximum, while some other finding are different in the sense that the seasonal variability in particle formation rate was quite modest and could not be established earlier (Nieminen et al., 2018). There is one more aspect which may be worth realising in this respect. A large fraction of compounds contributing to NPF and growth in cities can originate from anthropogenic precursors (Vakkari et al., 2015). Their emissions may peak any time of year depending on human habits and requirements (Nieminen et al., 2018). Nevertheless, the fact that our monthly distributions of the dynamic properties in urban environments follow the

universal summer maximum behaviour may indicate the overall prevailing role of atmospheric
photochemistry coupled with biogenic emissions of aerosol precursor vapours.

The monthly mean $J_6$, $GR_{10}$ and $H_2SO_4$ proxy data still have considerable uncertainty, which
makes their interpretation not yet completely conclusive. The uncertainties are influenced by
inherent fluctuations in the primary data sets, enhancing effects caused by combining some
individual primary data into compound variables (such as $H_2SO_4$ proxy), number of data items
available for different properties and months, variations in other or unknown relevant
environmental conditions, and by the variability in relative nucleation occurrence frequency
from year to year. The resulting uncertainties are expected to decrease with the length of the
available data sets, which emphasized the need to continue the measurements.

The monthly distributions of CS, and $SO_2$ and $NO_x$ concentrations could be represented by
constant values of the overall means and SDs of $(9.4\pm4.3)\times10^{-3}$ $s^{-1}$, $4.7\pm2.1$ $\mu g$ $m^{-3}$ and $81\pm38$
$\mu g$ $m^{-3}$, respectively with an acceptable accuracy. This suggests that these variables in Budapest
do not critically or substantially affect the dynamic properties (or the event occurrence).
Monthly distributions of air $T$ and $O_3$ concentration showed a maximum over summer months,
while RH reflected the $T$ tendency. In addition, monthly averages of $T$ on event days and on
non-event days were similar. Both higher biogenic emissions and typically stronger
photochemistry are expected during the summer, which enhance the production rate of
nucleating and condensing vapours, while there is practically nothing extra in the first
approximation (except for extreme $T$s) that would suppress the dynamical properties (Kerminen
et al., 2018). As result of these complex effects, the dynamic rates showed a summer maximum.
This is consistent with the results from other urban and non-urban studies (Nieminen et al.,
2018). Distribution of CO was more changing and without obvious tendentious temporal
structure or feature than for the other gases, and, therefore, its interpretation is encumbered so
far. However, it doesn't seem to substantially affect the dynamic properties.

Distributions of monthly average ratios of major variables on NPF event days to that on non-
event days for the joint city centre data set are summarised in Fig. 3. It is noted that the
differences in the number of non-event days and event days are the largest in winter and smallest
in spring (Fig. 1). The annual mean ratios for $N_{6-100}$, GRad, $SO_2$ and $O_3$ were above unity, for
$N_{100-1000}$ and RH, they were below unity, while the value of CS, $NO_x$ and CO were close to each
other on both types of days. Ultrafine particles are generated by NPF and growth processes in

a considerable amount; their concentration was larger by 23% on event days than on non-event days. This agrees with our earlier assessment of the NPF contribution as a single source of particles based on nucleation strength factor $NSF_{GEN}$ of 13% as a lower estimate (Salma et al., 2017). The other variables of the first group above represent conditions which favour atmospheric nucleation and particle growth, i.e., strong solar radiation, precursor gas and general photochemical activity, respectively. Particles in the size range of 100–1000 nm (the pre-existing particles with a relatively long residence time) express condensation and scavenging sink, which represents a competing process to nucleation. There is also evidence that RH acts against continental NPF process (Hamed et al., 2011).

It is also seen in Fig. 3 that NPF and growth events in winter took place preferably when $N_{100-1000}$, CS, RH, $NO_x$, and CO concentrations were especially low and $O_3$ concentration was unusually large. It can be explained by considering that the basic preconditions of NPF events are realised by the ratio of source and sink terms for condensing vapours. The source strength in winter is often decreased substantially in the Budapest area (Salma et al., 2017) due to lower solar radiation and less (biogenic) chemical precursors in the air. Nevertheless, NPF can still occur if the sink becomes even smaller. This also explains the relatively low event day-to-non-event day ratios for $N_{6-100}$ observed in winter months. Full exploitation of the data base by multistatistical and other methods has been in progress and is to be published in a separate article.

**Figure 3.** Distributions of ratios for monthly median concentrations of $N_{6-100}$, $N_{100-1000}$, $SO_2$, $O_3$, $NO_x$ and CO, and for monthly mean condensation sink for vapours (CS), global solar radiation (GRad), air temperature ($T$) and relative humidity (RH) on NPF event days to that on non-event days in the city centre for the joint 5-year long time interval. The horizontal lines represent annual mean ratios.

**4.3 Relationships**

Pearson's coefficients of correlation ($R$) between $J_6$ and $GR_{10}$ revealed significant linear relationship between them for all annual data sets (the mean $R$ and SD were 0.768±0.099, number of data pairs $n$=243). This confirms that formation of new aerosol particle and their growth to larger sizes are tightly and positively linked together. It should be noted that $J_6$ and $GR_{10}$ are not completely independent variables (see Eq. 1 and Table S1). The linear relationship

between the dynamic properties was observed under different atmospheric conditions in many
environments (Nieminen et al., 2018).

The dynamic properties can also be coupled to the concentrations of aerosol precursor
compounds and properties of a pre-existing particle population, thus to atmospheric
environment (Kerminen et al., 2018). It is, therefore, sensible to investigate the city centre and
near-city background data separately. Scatter plots between $J_6$ and $GR_{10}$ for the 1-year long
measurement time intervals are shown in Fig. 4. For the city centre, the regression lines follow
the line with a slope of 1 in all 5 years. The mean slope ($b$) with SD for the joint 5-year long
city centre data set was $b$=0.94±0.07 expressed formally in a unit of $cm^{-3}$ $s^{-1}$ $nm^{-1}$ h. At the
same time, the regression line for the near-city background deviated significantly with a
$b$=0.67±0.10 $cm^{-3}$ $s^{-1}$ $nm^{-1}$ h from the $J_6$ vs. $GR_{10}$ dependency for the city centre. This can
imply that NPF and growth processes advance in a different manner in these 2 environments.
This is likely related to the differences between the city and its close environment as far as the
atmospheric composition (for instance, the VOC and $NO_x$ concentrations), chemistry and
physics, and other delicate conditions are concerned (Paasonen et al., 2018). The narrower
range and smaller number of individual dynamic properties available for the near-city
background relative to those in the city centre represent some inherent limitation or weakness
in the explanation, and, therefore, it can strictly be regarded as a working hypothesis.

**Figure 4.** Scatter plots for aerosol particle formation rate $J_6$ and particle diameter growth rate $GR_{10}$ in
city centre (a and c–f) and near-city background (b) separately for the 1-year long measurement time
intervals. Number of data point ($n$), their coefficient of correlation ($R$) and the intercept ($a$) and slope
($b$) of the regression line with standard deviations are also indicated. The lines in black represent the
line with a slope of 1, the solid lines in red show the regression lines, while the dashed parts in red are
extrapolated from the regression line.

The intercepts ($a$) of the regression lines were identical for all data sets within their uncertainty
interval. The mean intercept and SD were estimated to be −1.7±0.8 $cm^{-3}$ $s^{-1}$. This finding is
interpreted as the existence of a minimum GR or more exactly of a minimally required GR that
leads to $J_6$>0. Particles that exhibit at least this level of GR can escape coagulation mainly with
larger particles and reach the detectable diameter (6 nm in our case) by condensational growth.
The minimal GR was derived as $GR_{min}$=−$a/b$, and its mean and SD are 1.8±1.0 $nm$ $h^{-1}$ for the
conditions ordinarily present in the Budapest air. Nucleation processes which are initiated under
circumstances that cause the newly formed particle with a diameter of 10 nm to grow with a
rate <$GR_{min}$ are normally not observed. Anyway, these are expected to be events with relatively
small $J_6$ (weak phenomena) due to the relationship between $GR_{10}$ and $J_6$. The events with GR
larger but close to this limit could be still masked by fluctuating experimental data. Their
identification and evaluation can be made feasible by decreasing the lower measurement
diameter limit of DMPS systems down to 3 nm, or by different instruments such as particle size
magnifier or neutral cluster and air ions spectrometer.

Correlations between individual $H_2SO_4$ proxy values on one side and $J_6$ or $GR_{10}$ on the other
side were not significant. This is consistent with the corresponding conclusion of Sect. 4.2 and
with the earlier results according to which the mean contribution of $H_2SO_4$ condensation to the
particle $GR_{10}$ was only 12.3% in Budapest (Salma et al., 2016b). The lack of correlation and
the average concentrations of $SO_2$ derived separately for event and non- event days suggest that
this precursor gas is ordinarily available in excess and, therefore, it is usually not the lack of
$SO_2$ gas itself, which limits the NPF and growth events in Budapest. Instead, the reaction rate
of oxidation of $SO_2$ to $H_2SO_4$ in the gas phase - likely governed by photochemical conditions -
, and other chemical species than $H_2SO_4$ can have larger influence on the particle growth. The
role of $H_2SO_4$ in the nucleation process and early particle growth could be still determinant or
relevant.

Coefficients of correlation between CS on one side and $J_6$ or $GR_{10}$ on the other side for the joint
city centre data sets were modest ($R$=0.41 and 0.32, respectively with $n$=194 and 197,
respectively). This is simply related to the fact that larger GR values are typical for polluted
urban air (Kulmala et al., 2017) since particles capable of escaping coagulation scavenging need
to grow faster in comparison to cleaner environments, and the enhanced requirements for the
growth are linked to increased formation rates as well. It should be noted here that the GR of
newly formed particles to larger sizes is primarily coupled to 1) CS, which is further linked to
the entire aerosol particle population (including the newly formed particles, thus the NPF itself),
2) to the total concentration and some physicochemical properties of non-volatile gaseous
compounds and 3) to their production rate in the gas phase from aerosol precursor compounds
(e.g. Kerminen et al., 2018). These couplings could result in rather complex behaviour, and
their understanding is essential when analysing atmospheric observations.

As far as the pollutant gases are concerned, no correlation could be identified between $J_6$ or
$GR_{10}$ on one side and the gas concentrations on the other side. The coefficients of correlation
between CS and $NO_x$ or CO were modest ($R$=0.37 and 0.42, respectively with $n$=164 and 152,
respectively), while correlation of $NO_x$ and CO on one side with WS was also modest but
negative ($R$= –0.32 and –0.42, respectively with $n$=167 and 155, respectively). The former
relationships can be explained by the fact that vehicular road traffic in cities is a considerable
and common source of $NO_x$, CO and primary particles (Paasonen et al., 2016), and the emitted
particles largely contribute to CS levels. The latter relationships are linked to the effect of large-
scale air mass transport (often connected to high WSs) on urban air pollution or air quality.

**4.4 Extreme and multiple events**

The data sets of $J_6$, $GR_{10}$ and $\Delta t$ containing all, 247 individual values each could be
characterised by lognormal distribution function. This is demonstrated by log-probability graph
for $J_6$ in Fig. S2 as example. The coefficient of determination, median and geometric standard
deviation for $J_6$, $GR_{10}$ and $\Delta t$ data sets were 0.990, 4.0 cm$^{-3}$ and 2.3; 0.993, 6.8 nm h$^{-1}$ and 1.46;
and 0.998, 02:57 (0.123 d) and 1.74, respectively. It is noted that the findings derived for the
separate city centre data set are very similar to the results presented above.

One of the major properties of this distribution type is that it contains relatively large individual
data with considerably high abundances. There were 5 individual $J_6$ and 5 individual $GR_{10}$ data
above the 98% percentile of the data sets, which belonged to 9 separate NPF and growth events
(days). Their specifications, properties and parameters are summarised in Table 3. All these
events occurred in the city centre from April to September. The medians of $J_6$, $GR_{10}$, CS and
air $T$ for the subsets of these 9 extreme event days were larger by factors of 5.2, 2.4, 1.5 and
1.4, respectively than for the city centre data. At the same time, the medians of the other
atmospheric properties and concentrations in these 2 respective data sets agreed within
approximately 10%. There was a single event associated with an extreme $H_2SO_4$ proxy (of
23×10$^5$ µg m$^{-5}$ W s) and relatively low $NO_x$ concentration (44 µg m$^{-3}$), which indicate
exceptionally favourable conditions for NPF and growth. In addition to this case, there were
only a few days that were characterised by an unusually large CS (23×10$^{-3}$ s$^{-1}$) – which could
in turn be linked to higher dynamic rates (Sect. 4.3) – or by somewhat larger $SO_2$ (8.1 µg m$^{-3}$)
or lower $NO_x$ concentration (34 µg m$^{-3}$). For all the other events, however, no simple or

compound property of the investigated variables could explain the extreme rates. Instead, they may be related to some other chemical species and/or atmospheric processes, which were not including in the present study.

**Table 3.** Date (in a format of dd–MM–yyyy), new particle formation rate $J_6$ (in a unit of $cm^{-3}\ s^{-1}$), particle diameter growth rate $GR_{10}$ ($nm\ h^{-1}$), starting time $t_1$ of nucleation (HH:mm UTC+1), duration time interval $\Delta t = t_2 - t_1$ of nucleation (HH:mm), mean condensation sink CS during the nucleation process ($10^{-3}\ s^{-1}$), daily maximum gas-phase $H_2SO_4$ proxy ($10^4\ \mu g\ m^{-5}$ W s), daily mean air temperature $T$ (°C), daily mean relative humidity RH (%), daily median concentrations of $SO_2$, $O_3$, $NO_x$ ($\mu g\ m^{-3}$) and CO ($mg\ m^{-3}$) gases, and the type of the onset for extreme quantifiable NPF and growth events. The cells in yellow indicate the values which are above the 98% percentile of the corresponding data sets. N.a.: not available.

| Date/ property | 15– 09– 2009 | 20– 04– 2014 | 19– 05– 2015 | 04– 07– 2015 | 28– 05– 2017 | 25– 06– 2017 | 02– 08– 2017 | 31– 08– 2017 | 09– 09– 2017 |
|---|---|---|---|---|---|---|---|---|---|
| $J_6$ | 15.9 | 17.8 | 24 | 16.3 | 27 | 33 | 30 | 47 | 17.3 |
| $GR_{10}$ | 17.4 | 19.0 | 12.2 | 18.0 | 9.2 | 17.0 | 11.8 | 21 | 19.8 |
| $t_1$ | 10:20 | 08:52 | 08:52 | 09:38 | 06:34 | 10:18 | 07:39 | 10:06 | 11:38 |
| $\Delta t$ | 01:23 | 01:42 | 03:57 | 02:06 | 07:15 | 02:46 | 06:58 | 06:19 | 02:06 |
| Proxy | 38 | 42 | 25 | 16 | 229 | 41 | 69 | 92 | 45 |
| CS | 13.4 | 8.9 | 13.7 | 11.9 | 6.9 | 10.5 | 23 | 18.2 | 15.5 |
| $T$ | 20 | 13.0 | 22 | 26 | 20 | 24 | 29 | 23 | 19.1 |
| RH | 60 | 62 | 48 | 40 | 40 | 68 | 49 | 47 | 58 |
| $SO_2$ | 6.1 | 2.5 | 4.4 | 2.3 | 3.4 | 3.1 | 5.6 | 8.1 | 6.6 |
| $O_3$ | 16.3 | 43 | n.a. | 33 | 61 | 56 | 34 | 24 | 12.9 |
| $NO_x$ | 69 | 34 | 174 | 70 | 44 | 66 | n.a. | 109 | 112 |
| CO | 0.42 | n.a. | 0.71 | 0.33 | 0.31 | 0.50 | 0.97 | 0.62 | 0.71 |
| Onset | ordinary | double | broad | ordinary | broad | broad | broad | broad | ordinary |

Each quantifiable NPF and growth event was labelled as ordinary or broad by visual inspection of its beginning part. If the width of the beginning was smaller than approximately 2 h or there was a determinant single growth curve (rib) on the size distribution surface plot then the onset was labelled as ordinary, otherwise as broad (Fig. S1b and S1c for broad onsets). Broad onsets can be generated by 1) long-lasting nucleation process, 2) disrupted and started over nucleation due to changing atmospheric and meteorological conditions or 3) multiple nucleation processes close to each other in time (Salma et al., 2016b). The broad onsets were specified as doublets if the nucleation mode could be separated into 2 submodes by size distribution fitting.

Approximately 40% of all quantifiable events had a broad onset. This indicates that events with
broad/multiple onsets are abundant in the urban environment, which could be an important
difference from remote or clean atmospheres.

For ca. 10% of all quantifiable event days, it was feasible to calculate 2 sets of dynamic
properties for onsets 1 and 2 with a reasonable accuracy. In the near-city background, the
medians of $J_6$ and $GR_{10}$ for the onset 1 were similar to the corresponding medians for the whole
near-city background data set, while for the onset 2, they were substantially larger, namely 4.1
$cm^{-3}\ s^{-1}$ and 10.0 nm $h^{-1}$, respectively (cf. Table 2). Actually, the latter values were closer to
the medians of the city centre than for the near-city background. Approximately 75% of the
doublets resulted in individual onset2/onset1 ratios larger than unity. Their overall median
ratios for $J_6$ and $GR_{10}$ were similar and approximately 1.2, while for the near-city background,
they were about 2. The results are in line with our earlier conclusion according to which the
second onsets (if it is a new formation process and not just a started over event) are more
intensive than the first onsets (Salma et al., 2016b). These particles also grow faster. This can
be explained by the fact that the first event is of regional scale since its dynamic properties
resemble those of the regional background (Yli-Juuti et al., 2009), while the later event can be
characterised by values typical for the city centre (Salma et al., 2016b). The later event (or
events) are mainly caused and governed by sub-regional processes. These findings are also
coherent with a previous observation of NPF and growth events with multiple onsets in semi-
clean savannah and industrial environments (Hirsikko et al., 2013), and they also fit well into
the existing ideas on mixing regional and urban air parcels that exhibit different properties such
as precursor concentrations, $T$ and RH (Kulmala et al., 2017).

**5 Conclusions**

Magnitude of the particle number concentration level produced solely by NPF and growth can
roughly be estimated by considering the median $J_6$, median duration of nucleation $\Delta t$ (their
distribution function is lognormal; Table 2) and the mean coagulation loss of these particles
$F_{coag}$ (0.17; Sect. 3.1 and Table S1) as: $J_6 \times \Delta t \times (1 - F_{coag})$. In central Budapest, it yields a
concentration of $10^4\ cm^{-3}$. This is in line with another result achieved by nucleation strength
factor (Salma et al., 2017). More importantly, the estimated concentration is comparable to the
annual median atmospheric concentrations (Table 1). This simple example indicates that the
phenomenon is relevant not only for aerosol load and climate issues on regional or global spatial
scales, which were first recognised. It is sensible also to study the effects of NPF and growth
events on urban climate and health risk for inhabitants since they produce a large fraction of
particles even in cities.

Similar recognitions have led to emerge of urban atmospheric nucleation studies. As part of this
international progress, we presented here a considerable variety of contributions, which became
feasible thank to gradually generating, multi-year long, critically evaluated, complex and
coherent data sets. Dynamic and timing properties of 247 NPF and growth events were studied
together with supporting aerosol properties, meteorological data and pollutant gas
concentrations in near-city background and city centre of Budapest for 6 years. The results and
conclusions derived form in important component that is based on atmospheric observations.
The present study can also be considered as the first step toward a larger and more
comprehensive statistical evaluation process.

Further dedicated research including sophisticated measurements, data evaluations and
modelling studies is required to find and identify additional chemical species and their
processes, and to account their multifactorial role in more detail. Such measurement campaign
focusing on chemical composition of molecular clusters, precursors and nucleating vapours by
applying recent expedient instruments in Budapest over the months of the highest expected
event occurrence has been just realised within a frame of an international cooperation. Its
perspective results can hopefully provide additional valuable information for some of the
conclusion base on indirect evidence for the time being and can further clarify the overall
picture on urban multicomponent nucleation and growth phenomenon.

*Data availability.* The observational data used in this paper are available on request from the
corresponding author or at the website of the Budapest platform for Aerosol Research and Training
(http://salma.web.elte.hu/BpART).

*Author contributions.* I.S. designed the study, performed most data analysis, interpreted the results and
wrote the paper. Z.N. performed most measurements and data treatment, and contributed to the data
analysis.

*Competing interest.* The authors declare that they have no conflict of interest.

*Acknowledgements.* The authors thank Markku Kulmala and his research team at the University of
Helsinki for their cooperation. Financial support by the National Research, Development and Innovation
Office, Hungary (contracts K116788 and PD124283); by the European Regional Development Fund
and the Hungarian Government (GINOP-2.3.2-15-2016-00028) is gratefully acknowledged.

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

 **Supplementary material**

 **Table S1.** Relative contributions of particle number concentration increment ($dN_{nuc}/dt=dN_{6-25}/dt-$
 $dN_{Ai,<25}/dt$), coagulation scavenging loss ($F_{coag}$) and growth out of particles from the diameter interval
 of 6–25 nm ($F_{growth}$) relative to the formation rate $J_6$ in the near-city background and city centre
 separately for 1-year long measurement time intervals. The measurement year and number of
 quantifiable NPF and growth events ($n$) are also shown.

| Environment and year/ statistics | Contribution in % | | |
|---|---|---|---|
| | $dN_{nuc}/dt$ | $F_{coag}$ | $F_{growth}$ |
| **Background, 2012–2013, $n$=43** | | | |
| Minimum | 45 | 4 | 2 |
| Maximum | 93 | 38 | 26 |
| Mean | 76 | 14 | 10 |
| St. deviation | 12 | 9 | 5 |
| **Centre, 2008–2009, $n$=31** | | | |
| Minimum | 32 | 13 | 3 |
| Maximum | 84 | 44 | 38 |
| Mean | 54 | 29 | 18 |
| St. deviation | 13 | 8 | 9 |
| **Centre, 2013–2014, $n$=48** | | | |
| Minimum | 43 | 9 | 3 |
| Maximum | 86 | 37 | 30 |
| Mean | 63 | 22 | 15 |
| St. deviation | 11 | 7 | 7 |
| **Centre, 2014–2015, $n$=56** | | | |
| Minimum | 45 | 6 | 2 |
| Maximum | 91 | 46 | 32 |
| Mean | 70 | 17 | 14 |
| St. deviation | 12 | 7 | 8 |
| **Centre, 2015–2016, $n$=17** | | | |
| Minimum | 50 | 4 | 2 |
| Maximum | 92 | 43 | 30 |
| Mean | 74 | 14 | 11 |
| St. deviation | 11 | 9 | 8 |
| **Centre, 2017–2018, $n$=52** | | | |
| Minimum | 44 | 4 | 3 |
| Maximum | 93 | 41 | 31 |
| Mean | 70 | 17 | 13 |
| St. deviation | 11 | 8 | 7 |

**Table S2.** Ranges, averages and standard deviations of condensation sink value during the nucleation
process, daily maximum gas-phase H$_2$SO$_4$ proxy, daily mean air temperature and daily mean relative
humidity on quantifiable NPF and growth events in the near-city background and city centre separately
for the 1-year long measurement time intervals and for the joint 5-year long city centre data set.

| Environment | Background | Centre | | | | | |
|---|---|---|---|---|---|---|---|
| Time interval | 2012–2013 | 2008–2009 | 2013–2014 | 2014–2015 | 2015–2016 | 2017–2018 | All 5 years |
| **Condensation sink, CS ($10^{-3}$ s$^{-1}$)** | | | | | | | |
| Minimum | 1.63 | 3.1 | 2.0 | 2.4 | 1.69 | 2.1 | 1.69 |
| Median | 5.6 | 9.5 | 9.9 | 8.6 | 5.0 | 8.4 | 8.9 |
| Maximum | 14.6 | 21 | 17.8 | 21 | 18.4 | 18.5 | 21 |
| Mean | 6.2 | 11.0 | 10.4 | 9.4 | 6.8 | 8.7 | 9.4 |
| St. deviation | 3.1 | 4.9 | 3.7 | 4.2 | 4.2 | 4.6 | 4.3 |
| **Gas-phase H$_2$SO$_4$ proxy ($10^4$ µg m$^{-5}$ W s)** | | | | | | | |
| Minimum | 40 | 10.9 | 12.2 | 5.8 | 34 | 7.3 | 5.8 |
| Median | 93 | 39 | 40 | 38 | 79 | 46 | 41 |
| Maximum | 163 | 96 | 139 | 135 | 190 | 134 | 190 |
| Mean | 93 | 39 | 45 | 42 | 82 | 50 | 48 |
| St. deviation | 32 | 17 | 27 | 23 | 38 | 31 | 29 |
| **Air temperature, $T$ (°C)** | | | | | | | |
| Minimum | −5.2 | −0.46 | −1.78 | −1.19 | −1.07 | 1.21 | −1.78 |
| Median | 11.5 | 17.1 | 16.8 | 15.3 | 14.2 | 16.7 | 16.1 |
| Maximum | 27 | 23 | 28 | 28 | 28 | 27 | 28 |
| Mean | 11.5 | 16.3 | 15.7 | 15.0 | 13.6 | 16.4 | 15.5 |
| St. deviation | 8.1 | 5.6 | 6.9 | 7.2 | 8.3 | 6.5 | 6.8 |
| **Relative humidity, RH (%)** | | | | | | | |
| Minimum | 41 | 32 | 41 | 31 | 39 | 36 | 31 |
| Median | 63 | 49 | 60 | 50 | 55 | 52 | 53 |
| Maximum | 91 | 74 | 78 | 77 | 89 | 73 | 89 |
| Mean | 64 | 51 | 60 | 50 | 56 | 52 | 54 |
| St. deviation | 12 | 11 | 10 | 9 | 12 | 9 | 11 |


**Table S3.** Ranges, averages and standard deviations of daily median concentrations of $SO_2$, $O_3$, $NO_x$
and CO gases on quantifiable NPF and growth event days in the near-city background and city centre
separately for the 1-year long measurement time intervals and for the joint 5-year long city centre data
set.

| Environment | Background | Centre | | | | | |
|---|---|---|---|---|---|---|---|
| Time interval | 2012–2013 | 2008–2009 | 2013–2014 | 2014–2015 | 2015–2016 | 2017–2018 | All 5 years |
| $SO_2$ concentration ($\mu g \, m^{-3}$) | | | | | | | |
| Minimum | 4.4 | 3.4 | 2.0 | 0.90 | 3.3 | 0.80 | 0.80 |
| Median | 6.2 | 5.3 | 5.1 | 3.9 | 5.2 | 3.7 | 4.8 |
| Maximum | 11.7 | 8.3 | 8.2 | 10.4 | 11.4 | 7.0 | 11.4 |
| Mean | 6.5 | 5.4 | 5.1 | 4.4 | 5.9 | 3.9 | 4.7 |
| St. deviation | 1.4 | 1.2 | 1.8 | 2.4 | 2.4 | 1.8 | 2.1 |
| $O_3$ concentration ($\mu g \, m^{-3}$) | | | | | | | |
| Minimum | 8.7 | 1.80 | 0.80 | 10.3 | 13.0 | 3.7 | 0.80 |
| Median | 61 | 44 | 25 | 35 | 36 | 29 | 31 |
| Maximum | 85 | 93 | 67 | 66 | 61 | 68 | 93 |
| Mean | 55 | 39 | 28 | 33 | 37 | 31 | 33 |
| St. deviation | 21 | 28 | 19 | 14 | 14 | 17 | 19 |
| $NO_x$ concentration ($\mu g \, m^{-3}$) | | | | | | | |
| Minimum | 4.9 | 13.0 | 34 | 32 | 30 | 17.8 | 13.0 |
| Median | 12.2 | 49 | 72 | 87 | 72 | 75 | 74 |
| Maximum | 66 | 213 | 143 | 186 | 120 | 167 | 213 |
| Mean | 15.8 | 62 | 77 | 96 | 76 | 79 | 81 |
| St. deviation | 12.1 | 42 | 28 | 41 | 24 | 33 | 38 |
| CO concentration ($mg \, m^{-3}$) | | | | | | | |
| Minimum | 0.167 | 0.26 | 0.30 | 0.26 | 0.29 | 0.20 | 0.198 |
| Median | 0.31 | 0.48 | 0.56 | 0.54 | 0.42 | 0.52 | 0.51 |
| Maximum | 0.87 | 0.76 | 0.79 | 0.95 | 0.88 | 0.86 | 0.95 |
| Mean | 0.38 | 0.47 | 0.54 | 0.55 | 0.46 | 0.51 | 0.52 |
| St. deviation | 0.18 | 0.13 | 0.14 | 0.16 | 0.16 | 0.15 | 0.15 |


**Figure S1.** Size distribution surface plots for NPF and consecutive particle diameter growth process as
banana-shape plots with an emission interference on 12–04–2015 (a), with limited growth of particles
on 19–03–2017 (b) and with a broad unresolvable onset on 01–04–2017 (c) in the city centre.

**Figure S2**. Log-probability graph of the formation rate $J_6$ and its cumulative frequency distribution for
$n$ individual data in the joint overall data set. The linear line in red represents the apparent fit to the data.
Coefficient of determination ($R^2$), median $J_6$ value ($M$) and its geometric standard deviation (GSD)
obtained from the fitted line are also shown.