# Peer review of "formation and consecutive growth events"

_Atmospheric Chemistry and Physics, 2018_

## Referee Comment (RC1) · Anonymous Referee #3 · 14 Jan 2019

An overview on 6 years of particle formation measurements in the urban area of Budapest is presented in this paper. The authors report particle formation rates at 6 nm (J6), growth rates at 10 nm GR10, starting time and duration of new particle formation events. They give yearly averages of J6, GR10, start and duration time as well as seasonal variations of these parameters together with event frequencies and further parameters like condensation sink (CS), temperature, humidity, O3, NOx, CO, SO2 and a sulfuric acid proxy. The authors show that the seasonal trends of event frequency and J6, GR10 do not coincide and that J6 and GR10 correlate more or less. From the latter they derive a lower limit of GR10, at which particle formation at 6 nm can still be seen. The authors do not find significant relations between J6 or GR10 with the sulfuric acid

proxy, CS and the gas concentrations. Finally, there is some discussion on extreme events, which is rather difficult to follow. The paper summarizes a large body of data and tries to extract information on the underlying processes of new particle formation. This is rather difficult as the lowest particle size they measure is 6 nm and growth rates can only be determined around 10 nm. The authors do not provide much more insight than in the paper of Niemienen et al., where they are coauthors of, except that the results are now based on a larger data set. Also the fact that the sulphuric acid proxy does not correlate with J6 and GR10 has already been reported in an earlier paper. Although sulphuric acid does only contribute 12.3% to GR10 it does not mean that it is not relevant for NPF (line 608). Many studies have shown a relation between NPF rates measured at small sizes and sulphuric acid, while the growth is dominated by organics. In Figure 4 the authors relate basically reciprocal (sulfuric acid proxy) versus reciprocal (sulfuric acid proxy) modulated by the GR. The linear relation is not surprising and does not lead to any conclusions. As the authors repeat several times in the paper NPF and growth is a complex process. Nevertheless, they test only relations of one single parameter with J6 or GR10. Why do the authors not make an attempt to combine parameters? It is known that low temperature stabilizes nucleating clusters and that organics promote growth and thus the survival probability. It might thus be worthwhile to look for a proxy representing condensing organics. I also question if daily averages are the appropriate parameter to inquire NPF mechanisms. Although it is worth to report on this large data set, I find the paper does not provide much new information and I do not see what the authors' "consequences of dynamic and timing properties" are as announced in the title. To be acceptable for ACP major improvements should be done. Besides the points mentioned above there are other issues. Line 151 and 494: What is the detection limit of the SO2 detector? Are the low SO2 concentrations measured significantly above DL? Line 318-319: I do not see a trend in particle concentrations. Table 2: the authors use local time as time base. We know that photolysis is an important driver of sulfuric acid and oxidant production. Would it not be more appropriate to use time after sunrise for starting time? Line 441: how can you

conclude that NPF is not sensitive to temperature? Indeed the yearly average does not vary much, but is the yearly average really important? What matters more is the temperature during an event in combination with formation rates of nucleating and condensing vapors. Line 498: What do you mean by "CO is less certain"? Figure 2: Is the low value of H2SO4-proxy in May real or an artefact? What is the reason for that? Line 545: This is not the line of equality. The units of each axis is different. There is also no discussion of this relation with respect to literature, e.g. Nieminen et al. Line 547: The difference between slopes for centre and near–city station is not very convincing. If the authors would also restrict the city centre plots to GR<10 nm/h I expect a large scatter of the slopes. The near-city data do not seem to be different from the other data. Line 559: It should say "that leads to J6>0". J=0 cannot be measured and is meaningless. Line 565: what do you mean by "weak phenomena"? Line 611ff: This explanation is unclear. Surely, GR need to be faster in urban areas but that does not mean that there could be no correlation. Simply speaking higher CS should lead to lower GR. Apparently, a positive correlation is found, isn't it? This would be counterintuitive. Section 4.4 needs much improvement. Line 739: Where does this number of contribution of NPF to total particle concentration come from? How was the analysis done?

---

## Referee Comment (RC2) · Anonymous Referee #4 · 14 Jan 2019

The manuscript presents a data set of particle size distribution by 5 years DMPS measurements at Budapest city center and 1 year measurement upwind of the city. Analysis of new particle formation is presented by particle formation rate J6 and growth rate GR10 as well as starting time and duration time interval. Factors affecting NPF are explored by relating to gas-phase H2SO4 proxy, condensation sink (CS), meteorological data, and concentrations of SO2, O3, NOx and CO. Despite there is no measurement for sub-6 nm particle and potential precursors for NPF, it is still an interesting data set could potentially contribute to a better understanding towards urban aerosols and constrain the atmospheric model. However, more detailed explore into the data would still be needed apart from performing correlation test between single parameters and

conclude there is no correlation.

Specific comments:

Page8, line 256: the mean new-to-old rate ratios of J6 were 1.23 for city center and 1.20 for near-city background. I would expect that traffic emission causes overestimation of formation rates because it is a source of nanoparticles. Please specify why correcting traffic emission in formation rates calculation gives higher J6.

Section 4.2: Discussion on NPF events frequency should include conditions of NPF days as well as non-NPF days. Properties discussed in the section are only based on events days. This could be misleading because non-events day conditions are not discussed. Line 484 conclude gas-phase H2SO4 are unlikely to be the limiting factor of NPF occurrence in Carpathian Basin including Budapest from the misalignment between the monthly occurrence frequency and the other properties. To make this statement solid, H2SO4 proxy for events days and non-events days is needed.

Page18, line548: Direct compare the numbers of J and GR or saying something contribute equally to the formation of particle and to their growth don't make sense because they are different physical variables. Correlation between J and GR are expected but comparison of the regression line with J6=GR10 doesn't give any useful information.

Page20: Lacking correlation with single parameters to J/GR doesn't tell too much as NPF is controlled by multiple parameters. With the size of the data set, authors could perform analysis on subsets of the data with certain constrains like temperature or H2SO4 proxy.

Page 20, line 625 to 636 and figure 4: GR/H2SO4 proxy =b*( 1/H2SO4 proxy)+a is equivalent to a*H2SO4 proxy+b=GR. A negative 'a' means the higher H2SO4, the lower the GR. This is contradictory to the interpretation of increasing gas-phase H2SO4 related to larger contribution of other vapors to particle growth. Another concern would be special care should be taken when combine H2SO4 proxy at sub-urban site and

urban site as the VOCs and NOx condition could be totally different but not taken into consideration.

Page 24, line 739: To make the full potential of the data set, more detailed studies on the contribution of NPF to regional particle concentration could be performed.

Spelling:

Line 113: mean see level-> mean sea level

Line 751: cloud -> CLOUD

---

## Author Comment (AC1) · 7 Feb 2019

**Response to Referee number 3**

7 February 2019

The authors thank Referee #3 for his/her work. We have considered all comments thoroughly and profoundly. Unfortunately, many of them cannot be accepted at all or fully. Our specific responses are as follows, while the textual modifications are highlighted in red or by crossing out in the revised MS.

The paper summarizes a large body of data and tries to extract information on the underlying processes of new particle formation. This is rather difficult as the lowest particle size they measure is 6 nm and growth rates can only be determined around 10 nm.

1.  The lower measurable particle diameter limit of DMPS/SMPS systems is important for identification of NPF and growth events and further data treatment. Evaluations of this type of atmospheric measurements are mostly based on particle diameter range <20 nm (e.g. Kulmala et al., Nat. Protoc., 7, 1651–1667, 2012). In order to separate reliably the NPF and growth events from huge emission peaks which can occur in cities and which can temporarily influence the size intervals down to even smaller diameters, it is highly preferable to have the lower limit below 10 nm. Our limit value of 6 nm was proved to be already satisfactory since it allows to identify and separate different particle generation processes (see e.g. Fig. S1b of the present MS and Salma et al., Atmos. Chem. Phys. 16, 7837–7851, 2016). It is also worth mentioning that from 6 urban cites involved in a recent global analysis of NPF over long-term measurements (Nieminen et al., Atmos. Chem. Phys., 18, 14737–14756, 2018), the lower diameter limit was 3 nm at 2 sites, it was 6 nm at 3 locations, while it was 11 nm at 1 of the sites, and both the $J_{nuc}$ and GR were determined for the diameter interval of 10–25 nm. All these indicate that in atmospheric studies, our experimental systems and evaluation protocols seem completely adequate for the time being.

The authors do not provide much more insight than in the paper of Niemienen et al., where they are coauthors of, except that the results are now based on a larger data set.

2.  The goals of the paper mentioned in the comment were largely different from our aims. We can list several important insights explicitly as examples which are part of the present MS and which were not dealt with in the referred paper. They primarily include 1) the evaluation

and discussion of monthly distributions of $J_6$ an $GR_{10}$ together with their relationships with nucleation occurrence frequency and relevant atmospheric parameters, 2) timing properties of NPF and growth events, 3) refinements of $J$ and GR calculations dedicated to urban environments, 4) statistical distributions of $J_6$ an $GR_{10}$, 5) occurrence and properties of extreme events and events with broad onset. These items represent a considerable piece of novelty and new knowledge. Furthermore, the results and conclusions are based on 247 quantifiable NPF and growth events in an urban environment, which means a rather strong background. Finally, we can quote from the Summary and conclusions section of the Nieminen et al. paper (p. 14750): "For future studies, it would be very valuable to make detailed investigations on the interdependencies among $J_{nuc}$, GR, and NPF event frequency, at both single measurement sites and among sites of seemingly similar environmental characteristics." This is exactly what we did in our MS. In addition to our arguments, we can offer to all persons involved a recent and excellent review paper of Kerminen et al., Environ. Res. Lett. 13 (2018) 103003, 2018 dedicated to field observations, which also gives a scientific outlook and summarizes future research needs, and which can help putting our present results and conclusions more adequately into a scientific frame of international atmospheric NPF and particle growth studies.

Also the fact that the sulphuric acid proxy does not correlate with J6 and GR10 has already been reported in an earlier paper. Although sulphuric acid does only contribute 12.3% to GR10 it does not mean that it is not relevant for NPF (line 608).

3. This conclusion was mentioned in the MS as a minor outcome of the study with the purpose of confirming earlier results (as explicitly stated in the line specified in the comment). The related sentence was modified now to emphasize the key role of $H_2SO_4$ in the nucleation process and early particle growth.

Many studies have shown a relation between NPF rates measured at small sizes and sulphuric acid, while the growth is dominated by organics. In Figure 4 the authors relate basically reciprocal (sulfuric acid proxy) versus reciprocal (sulfuric acid proxy) modulated by the GR. The linear relation is not surprising and does not lead to any conclusions. As the authors repeat several times in the paper NPF and growth is a complex process. Nevertheless, they test only relations of one single parameter with J6 or GR10. Why do the authors not make an attempt to combine parameters?

4. Figure 4 and the related discussion were removed from the MS to avoid any misunderstanding or incompleteness. The remaining part was also restructured, split into shorter pieces and clarified. Evaluation of the overall data set by multistatistical methods is indeed planned. This comprehensive evaluation is, however, to be accomplished after some markers or proxies for biogenic emission sources (such as e.g. photosynthetical activity) are also included. The extension of the present MS by this comprehensive statistical analysis would not fit among the present objectives and would not be advantageous or feasible considering both the length and timing of this MS as well. See also response no. 5.

It is known that low temperature stabilizes nucleating clusters and that organics promote growth and thus the survival probability. It might thus be worthwhile to look for a proxy representing condensing organics.

5. Chemical species including organics participating in the urban atmospheric NPF and growth were investigated in an intensive international measurement campaign in Budapest over March-May 2018 by deploying API TOF-MS with/without CI, PSM, AIS and DMPS systems. Some potential proxy values for condensing organics are under evaluation. This was mentioned in the Conclusions section, and it is further emphasized and explained in the revised version.

I also question if daily averages are the appropriate parameter to inquire NPF mechanisms. Although it is worth to report on this large data set, I find the paper does not provide much new information and I do not see what the authors' "consequences of dynamic and timing properties" are as announced in the title.

6. Daily averages were calculated for those variables which change slowly over a day (e.g. $[SO_2]$). For some other variables such as CS, we constrained the averaging for the time intervals from $t_1$ to $t_2$, thus over the nucleation process itself. Some other variables, such as the gas-phase $H_2SO_4$ proxy, were characterized by their daily maximum. They are accurate

specified in Section 4.1 Ranges and averages. As far as the novelty of the MS is concerned, we must refer again to the list in response no. 2. The main conclusions drawn from the dynamic and timing properties are readily collected in the Abstract.

Line 151 and 494: What is the detection limit of the SO2 detector? Are the low SO2 concentrations measured significantly above DL?

7.  The limit of determination (LOD) of the $SO_2$ analyzer system applied is approximately 0.2 µg m$^{-3}$. More than 98% of the hourly-mean concentrations were above the LOD. The information is also included into the text now.

Line 318-319: I do not see a trend in particle concentrations.

8.  The annual medians for the city centre in the measurement years 2008–2009, 2013–2014, 2014–2015, 2015–2016 and 2017–2018 are as follows: $11.5 \times 10^3$, $9.7 \times 10^3$, $9.3 \times 10^3$, $7.5 \times 10^3$ and $10.6 \times 10^3$ cm$^{-3}$, respectively. The first 4 data indicate unambiguously a decreasing tendency, while the last data point may look somewhat different. Rigorous statistical evaluation of the joint data set of particle number concentrations in various size fractions over a decennial time interval from 03–11–2008 to 02–11–2018 is in progress, and its preliminary results in the one hand, confirm the decreasing tendency, and in the other hand, reveal some fine structure to this dependency. This information was added to the revised MS.

Table 2: the authors use local time as time base. We know that photolysis is an important driver of sulfuric acid and oxidant production. Would it not be more appropriate to use time after sunrise for starting time?

9.  The suggestion represents an option, which can be consider for specific studies. In the present MS, we selected the local time as the time base of most data on purpose and as a compromise because we had experienced in several earlier investigations (e.g. Salma et al., Atmos. Environ., 92, 154–161, 2014) that it is the daily activity time pattern of inhabitants that substantially influences or determines many atmospheric sources and important processes in Budapest. It was explained in lines 123–125 of the original MS, and a reference

for the statement was included as well. The timing parameters of the NPF were given in UTC+1.

Line 441: how can you conclude that NPF is not sensitive to temperature? Indeed the yearly average does not vary much, but is the yearly average really important? What matters more is the temperature during an event in combination with formation rates of nucleating and condensing vapors.

10. The sentence mentioned was replaced from its original location to section 4.2 Monthly distributions. It was largely corrected and extended to a discussion by involving the temperature profiles on nucleation and non-nucleation days, biogenic emissions, photochemistry and results from other international studies.

Line 498: What do you mean by "CO is less certain"?

11. The related sentence was modified to express our intention better that the variability of CO was without obvious tendentious temporal structure or feature.

Figure 2: Is the low value of H2SO4-proxy in May real or an artefact? What is the reason for that?

12. It is the monthly distribution of daily maximum gas-phase $H_2SO_4$ proxy that is shown in Fig. 2. The mean value for May represents 23 days. Its low value seems to be influenced by enhanced effect of multiplying relatively low GRad with relatively small $[SO_2]$ for a few days particularly in 2015 (which was a strange year as far as the monthly distribution of nucleation frequency as well is concerned; see Fig. 1 of this response). The reliability of the monthly data is to be increased with the length of the overall data sets in the future. This additional information is added now in a synthetized manner to the text.

[Figure]

Figure 1. Monthly distribution of relative nucleation frequency in Budapest for measurement years of 2008–2009, 2012–2013, 2013–2014, 2014–2015, 2015–2016 and 2017–2018. The horizontal lines indicate the annual mean frequency. More information is given in the MS.

Line 545: This is not the line of equality. The units of each axis is different. There is also no discussion of this relation with respect to literature, e.g. Nieminen et al.

13. We used the expression "line of equality" in its broader sense, hence when the abscissa and ordinate are on the same scale even they do not have the same units. To the explicit request of Referee #3, however, we can change it to another expression, e.g. "line with a slope of 1". We also amended the discussion of the relationship between *J* and GR at several places by considering the international results available in the literature.

Line 547: The difference between slopes for centre and near–city station is not very convincing. If the authors would also restrict the city centre plots to GR<10 nm/h I expect a large scatter of the slopes. The near-city data do not seem to be different from the other data.

14. We were aware of this inherent limitation mainly caused by smaller dynamic properties (and partly by shorter measurement time interval) in the near-city background than in the city centre, and expressed it by ourselves in lines 554–556 of the original MS. Now, we reformulated the statement completely and turned it from a conclusion into a working hypothesis because a rigorous statistical treatment would indeed require stronger/larger variability in the near-city background data.

Line 559: It should say "that leads to J6>0". J=0 cannot be measured and is meaningless.

15. The suggestion was accepted and adopted.

Line 565: what do you mean by "weak phenomena"?

16. The related sentence was modified to express that we mean the NPF events with relatively small particle formation rate (weak events).

Line 611ff: This explanation is unclear. Surely, GR need to be faster in urban areas but that does not mean that there could be no correlation. Simply speaking higher CS should lead to lower GR. Apparently, a positive correlation is found, isn't it? This would be counterintuitive.

17. The GR of newly formed particles to larger sizes is primarily coupled to 1) CS, which is further linked to the entire aerosol particle population (including the newly formed particles, thus the NPF itself), 2) to the total concentration and some physicochemical properties of non-volatile gaseous compounds and 3) to their production rate in the gas phase from aerosol precursor compounds (e.g. Kerminen et al., Environ. Res. Lett. 13 (2018) 103003, 2018). Understanding these couplings is essential when analyzing atmospheric observations. It is not fully plausible to make intuitive expectations on simplified paired relationships, for instance between CS and GR, under such complexity. Therefore, we stuck to the experimental data and are to contribute to the phenomenological picture on the system of relationships in this part of the MS, which will be eventually leading to a comprehensive and

qualitative explanation of the connections in the future. We extended the sentence briefly with these additional arguments and explanation.

Section 4.4 needs much improvement.

18. We split the section into shorter parts and clarified it by clearer formulations.

Line 739: Where does this number of contribution of NPF to total particle concentration come from? How was the analysis done?

19. Typical number of particles generated by an NPF and growth event on a nucleation day was roughly estimated by considering the median $J_6$ and median duration of nucleation, $\Delta t$ (their distribution function is lognormal; see Table 2) and mean relative coagulation loss, $F_{coag}$ (see Table S1) as: $J_6 \times \Delta t \times (1-F_{coag}) = 4.6 \times 180 \times 60 \times 0.83 = 41 \times 10^3$ cm$^{-3} \approx 10^4$ cm$^{-3}$. This concentration is in line with other results achieved by nucleation strength factor according to which the particle number concentration due to NPF and growth process on a general nucleation day is increased by a factor of approximately 2 (Salma et al., Atmos. Chem. Phys., 17, 15007–15017, 2017). A more detailed description of the estimation process and the mathematical expression utilized are added now together with the last reference mentioned.

In addition to the issues above, we also adopted some smaller changes and added a few recent papers as references to further improve the MS.

Finally, we think that the comments of Referee #3 eventually helped us to formulate our thoughts and ideas better. We appreciate this. We wish, however, to emphasize that the major message of the MS lies in a considerable variety of contributions to the emerging research field of urban atmospheric NPF and growth, which have been becoming possible and increasingly recognized thank to gradually generating, several-year long, semi-continuous, critically evaluated, complex and coherent data sets. We further stressed this aspect of the MS now in the Conclusion section and added a new opening sentence to the Abstract as well.

Imre Salma

---

## Author Comment (AC2) · 7 Feb 2019

**Response to Referee number 4**

7 February 2019

The authors thank Referee #4 for his/her detailed, expertise and valuable comments to further improve and clarify the MS. We have considered all recommendations and made appropriate alterations. Our specific responses to the comments are as follows, while the detailed textual modifications are highlighted in red or by crossing out in the revised MS.

Page8, line 256: the mean new-to-old rate ratios of J6 were 1.23 for city center and 1.20 for near-city background. I would expect that traffic emission causes overestimation of formation rates because it is a source of nanoparticles. Please specify why correcting traffic emission in formation rates calculation gives higher J6.

1. Several modifications were simultaneously adopted in the revisited and refined calculations protocol of the new set of $J_6$ for the measurement years of 2008–2009 (city centre) and 2012–2013 (near-city background). They include the subtraction of particle number concentrations emitted by road traffic from $N_{6-25}$, which usually leads to a decrease in the coagulation loss and loss due to growth out from the diameter range of 6–25 nm, and which can sensitively influence the slope of the concentration change in time ($dN_{6-25}/dt$) in a positive or negative manner depending on the actual time evolution of perturbing emission source. In addition to that, the time interval in which this slope is considered to be constant was determined within a completely new treatment. We would also like to mention the mean relative contribution of the concentration increment, coagulation loss and growth out from the diameter interval to $J_6$ have different weights of 71%, 17% and 12%, respectively (lines 246–249 of the original MS, and Table S1). Furthermore, $J_6$ itself also depends on $GR_{10}$, which makes the relationships even more complex. These explain why the overall effect of urban influence generally resulted in increased dynamic properties. The mean new-to-old ratio for $J_6$ was larger for the city centre (1.23) than for the near-city background (1.20). It should also be emphasized that the re-calculation mainly affected the individual dynamic properties with relatively small absolute values. The whole process is considered as a methodological improvement over the years of research. The MS was amended by a more detailed description of Equation (1) and by a brief explanation of the issues above.

Section 4.2: Discussion on NPF events frequency should include conditions of NPF days as well as non-NPF days. Properties discussed in the section are only based on events days. This could be misleading because non-events day conditions are not discussed.

2. Information on the average CS (calculated for whole days), gas-phase $H_2SO_4$ proxy, GRad, air $T$ and concentration of some criteria pollutants on non-nucleation days were partly included now. Many properties are, however, biased by the seasonal cycle of solar electromagnetic radiation via the seasonal variation of new particle formation frequency, and therefore, they interpretation needs special attention. They are to be fully utilized and explained for investigating the changes in annual patter of relative nucleation frequency over the years, and a more comprehensive evaluation and discussion is to be realized in a future study outlined in response no. 5.

Line 484 conclude gas-phase H2SO4 are unlikely to be the limiting factor of NPF occurrence in Carpathian Basin including Budapest from the misalignment between the monthly occurrence frequency and the other properties. To make this statement solid, H2SO4 proxy for events days and non-events days is needed.

3. Averages of several atmospheric properties involved in the $H_2SO_4$ proxy were derived separately for the event days and non-event days, their effects were briefly discussed, and as a result of it, the statement mentioned was removed.

Page18, line548: Direct compare the numbers of J and GR or saying something contribute equally to the formation of particle and to their growth don't make sense because they are different physical variables. Correlation between J and GR are expected but comparison of the regression line with J6=GR10 doesn't give any useful information.

4. The sentence was modified to express clearly that we mean that the chemical species available in the air affect the formation rate and growth rate differently at the 2 urban sites. This could partially be caused by differences in chemical composition. We reformulated the whole statement completely and turned it from a conclusion into a working hypothesis because a rigorous statistical treatment would indeed require larger variability in the near-city background data.

Page20: Lacking correlation with single parameters to J/GR doesn't tell too much as NPF is controlled by multiple parameters. With the size of the data set, authors could perform analysis on subsets of the data with certain constrains like temperature or H2SO4 proxy.

5.  Evaluation of the overall data set by multistatistical methods is indeed planned. This comprehensive evaluation is, however, to be accomplished after some markers or proxies for biogenic emission sources (such as e.g. photosynthetical activity) are also included. The extension of the present MS by this comprehensive statistical analysis would not fit among the present objectives and would not be advantageous or feasible considering both the length and timing of this MS as well. The present study can be considered as the first step in a larger statistical evaluation process and which supplied orienting ideas on the specific directions to proceed in. This perspective further study is very briefly mentioned in the Conclusions section now.

Page 20, line 625 to 636 and figure 4: GR/H2SO4 proxy =b*( 1/H2SO4 proxy)+a is equivalent to a*H2SO4 proxy+b=GR. A negative 'a' means the higher H2SO4, the lower the GR. This is contradictory to the interpretation of increasing gas-phase H2SO4 related to larger contribution of other vapors to particle growth. Another concern would be special care should be taken when combine H2SO4 proxy at sub-urban site and urban site as the VOCs and NOx condition could be totally different but not taken into consideration.

6.  Figure 4 and the related discussion were removed from the MS to avoid any misunderstanding or incompleteness. The remaining part of the section was restructured, split into shorter pieces and clarified.

Page 24, line 739: To make the full potential of the data set, more detailed studies on the contribution of NPF to regional particle concentration could be performed.

7.  We fully agree on this item and will proceed in that direction in the future.

Spelling

Line 113: mean see level-> mean sea level
Line 751: cloud -> CLOUD

8.  The typing errors were corrected.

In addition to the issues above, we also adopted some smaller changes and added a few recent papers as references to further improve the MS.

Imre Salma

---

## Author Response (AR2)

**Response to Referee number 3**

April 2019

The authors thank Referee #3 for his/her expertise and valuable comments to further improve and clarify the MS. We have considered all recommendations and made appropriate alterations. Our specific responses to the comments are as follows, while the detailed textual modifications were amended in the marked-up version of the MS ver. 3.

The authors claim" daily activity time pattern of inhabitants determine many atmospheric sources and important processes". However, they choose UTC+1 throughout the year to characterize the starting time of NPF. In addition, Figure 4 clearly shows that radiation has an important impact on the particle formation and growth process. All this is contradictory to their statement. With their approach they find that "nucleation ordinarily starts at 09:15 UTC+1". I doubt that this result is very helpful for e.g. modelers, if they know that nucleation can happen sometime between 6 − 12 a.m. Of interest would be some information on the underlying process/drivers.

1. It is the direct emissions mainly from vehicular road traffic, some household activities and service sectors that follow the daily activity time pattern of inhabitants (Paasonen et al., ACP, 16, 6823–6840, 2016). These sources determine many atmospheric processes in cities in general. Local time (LT) scale is often used for them. We must stress the role of meteorology as well. Urban atmospheric NPF and growth events take place in this dynamic atmospheric environment, and they are associated with several precursors of both anthropogenic and biogenic origin, further secondary chemical species and meteorological conditions. Since the impact of GRad seems to be important and biogenic emissions may also strongly influence the whole process, and since both these effects are influenced by solar cycling, we expressed the starting time parameter $t_1$ of NPF in UTC+1. This also facilitates its descriptive statistics and its comparison among different environments. (In LT, this would be misleading.) We do not see contradiction here. It is not completely clear where the interval of 06:00–12:00 came from; the NPF events in Budapest (as representative of a large Central European city) start between 08:00 and 10:50 UTC+1 in the centre, and between about 07:10 and 09:30 UTC+1 in the near-city background with average values given in Table 2, and there is a delay of about 1 h in urban NPF with respect to its close background. This could interest modellers.

Line 266-270: this is an incomplete sentence. Probably you need to delete "and which"

2.  The sentence mentioned was split into 2 parts to clarify its exact meaning and to help more fluent reading.

Line 274-275: These explain….. This sentence is not clear.

3.  The sentence was improved by an explicit grammatical subject.

Line 318-319: the factor could distort the dynamic relationships …. It could also be that neglecting this factor leads to a bias. It can be either way.

4.  The part of the sentence was removed. These factors are basically related to the limitations of the proxy value.

Line 506-508: this sentence is unclear. Why should there be a seasonal bias? I assume you compare event and non-event days per month. Otherwise it does not make sense.
Line 508: I do not see a seasonal cycle for CS.

5.  The sentences mentioned were extended with further aspects to clarify their meaning.

Line 526: "uncertainty", should this not be variability?

6.  We referred to the standard deviations of the monthly mean values, which are expected to become smaller as the length of the data set gets larger.

Line 542: "higher biogenic emissions and typically stronger photochemistry are expected" higher photochemical activity also enhances formation of nucleating species from anthropogenic VOCs. It must not necessarily be biogenic species.

7.  The sentence was reformulated to clarify that we originally meant 2 processes of 1) higher biogenic emissions and 2) stronger photochemistry as separate contributors. The latter process enhances the production of nucleating chemical species from both biogenic and non-biogenic (e.g., anthropogenic) sources.

Line 544: "nothing extra that would suppress the dynamical properties". This is not fully correct. Higher T in summer leads to higher vapor pressure and decreases supersaturation. This decreases nucleation and growth rates.

8. The sentence (more precisely "practically nothing extra…") was cited from Kerminen et al., 2018, and the effect of $T$ mentioned was taken into account as an ordinary impact. We modified the formulation to clarify undergoing processes in more detail.

Line 666: In line 467 you estimate H2SO4 to 5E05 cm-3. 12.3% of growth by H2SO4 results in about 1 nm/h. This would need about 5E07 H2SO4. How do you explain this discrepancy of 2 orders of magnitude?

9. It is not fully straightforward to follow the requirement of the $H_2SO_4$ concentration of $5\times10^7$ cm$^{-3}$ from the comment. The difference could be caused by the scaling factor as discussed in lines 316–320 of the MS ver. 2. Atmospheric measurements of $H_2SO_4$ by an CI APi-TOF MS on the spot in last spring also indicate values around $7\times10^5$ cm$^{-3}$.

Line 668: what do you mean with "excess"? "formation of H2SO4 is likely governed by photochemical conditions": what other processes than photochemical production could do it under the given urban conditions?

10. We utilized the expression "in excess" in its chemical sense, thus, meaning that it is usually not the lack of $SO_2$ gas itself, which limits the NPF and growth events in Budapest, and that the conditions for photochemical oxidation of $SO_2$ to $H_2SO_4$ (its reaction rate in the gas phase) could be one of the governing process. The related sentences were modified.

Line 794: replace cloud by CLOUD. You need to say what this is.

11. We referred to cloud chamber experiments in general including likely the most outstanding facility of CLOUD at CERN. The sentence was removed for another reason.

The writing of the text is still quite complicated and not so concise. English needs substantial improvement yet.

12. Several sentences were split into shorter parts. The writing and language of the MS was improved.

Imre Salma

**Response to Referee number 5**

April 2019

The authors thank Referee #5 for his/her work. The Referee's report is very short, and it contains only 1 real issue. This is, unfortunately, based on a misunderstanding of the objectives of the MS. Our responses are as follows, while the detailed textual modifications were amended in the marked-up version of the MS ver. 3.

The dataset presented consist of several years of urban aerosol dynamics data. However, the data analysis presented is too simple – only average values, standard deviations and linear regressions are discussed, and the uncertainties are too large to draw robust conclusions in many cases. Overall, the authors should consider a different approach to the data analysis in order to draw interesting and substantial conclusions from the dataset.

1. In the present MS, we mainly focused on various aspects of dynamic and timing properties. The objectives include for instance: 1) the evaluation and discussion of monthly distributions of $J_6$ an $GR_{10}$ together with their relationships with nucleation occurrence frequency and relevant atmospheric parameters, 2) timing properties of NPF and growth events, 3) refinements of $J$ and GR calculations dedicated to urban environments, 4) statistical distributions of $J_6$ an $GR_{10}$, 5) occurrence and properties of extreme events and events with broad onset. These items represent considerable novelty and new knowledge.

   The objectives do not include the data analysis raised by the Referee. The overall extent of the data base available by now is estimated to contain critically evaluated records/lines for 7 measurement years, which each consists of size distribution data in 27 channels with a time resolution of ca. 8 min, particle number concentrations for 4 different size fractions ($N_{6-25}$, $N_{6-100}$=UF, $N_{100-1000}$ and $N$), 2 derived compound properties (CS and $H_2SO_4$ proxy), 5 meteorological data ($T$, RH, WS, WD and GRad with a time resolution of 10 min), concentrations of 5 pollutant gases ($SO_2$, NO, $NO_x$, $O_3$ and CO with a time resolution of 1 h) and attribution indices on the nucleation classes and workdays/holidays. Its comprehensive analysis requires specific and careful adaptation of multivariate statistical methods. We have been working on this in a separate project in cooperation with dedicated mathematical statisticians, and its results and conclusions are to be published in a new MS later.

Additionally, it does not make sense to mix a 5-year-long dataset for the city center with a 1-year-long dataset for the near-city site into the same data analysis.

2.  The NPF and growth processes in the city centre and near-city background environments were almost exclusively evaluated separately (see Tables 1 and 2, Figs. 2 and 3 for summary). We showed previously (Salma et al., ACP, 2016b) that the NPF events observed in the city centre of Budapest and its background usually happen above a larger territory in the Carpathian Basin as a spatially coherent and joint atmospheric phenomenon. From this aspect only and in selected specific cases such as the relative occurrence frequency distribution, the data sets for the centre and near-city background were joined and treated together in the first approximation. This was explicitly stated and argued for (lines 359–363 of the MS ver. 2). Nevertheless, we evaluated the average frequency distributions separately as well and found no substantial or tendentious differences in the distributions averaged for the overall data set, city centre data set and near-city background data set. This information was added to the MS ver. 3.

Furthermore, the title does not match the contents of the manuscript.

3.   The title was changed to express the content of the MS more closely.

In summary, we were able to find a positive message in this comment. We can fully agree with the Referee on the importance and necessity of further complex statistical analysis, and we regard the comment as an initiation or confirmation that we should continue in this way. At the same time, the major research results of the present MS (such as the items 1–5 listed above) should not be neglected and are to be considered as well when making the final decision.

Imre Salma

**Response to Referee number 6**

April 2019

The authors thank Referee #6 for his/her detailed, expertise and valuable comments to further improve and clarify the MS. We have considered all recommendations and made appropriate alterations. Our specific responses to the comments are as follows, while the detailed textual modifications were amended in the marked-up version of the MS ver. 3.

The title must be changed reflecting the real scope of the study.

1. The title was changed to express the content of the MS more closely.

Conclusions should be stated taking into account the limitations of the method employed. Some of sentences stated in the abstract ("The 15 NPF and growth events produce 4.6 aerosol particles with a diameter of 6 nm in 1 cm3 of air in 16 1 s…") and in the conclusion sections must be modified taking into account the limitations of the methodology and the scope of the study. The conclusion section does not exactly reflect the results obtained during this study. Thus, the present study did not provide information about effect of NPF in urban climate and health risk (l774-776). Cited form text (L792-793): "the present research…provided evidence that some important chemical players in the NPF and growth events are still missing". This cannot be considered as a conclusion obtained in this paper. The methodology used does not permit to identify key chemical players because the methods used are not the most adequate to this end.

2. The sentences cited from the Conclusions were meant as an outlook for further directions. To avoid any misunderstanding, we removed or modified substantially them. In addition, several other statements were reformulated into more modest expressions to insist more on the direct implications. In addition, the methodological limitations of the study were emphasized in the body of the MS ver. 3.

The statically treatment can be improved. The analysis should be performed for both the NPF and non-NPF events. Thus, in Section 4-2 and Figure 2, the analysis of the monthly evolution is limited to the NPF formation events. It should also be performed for the non-NPF events. A figure about daily evolution of NPF events, and other variables, could help for interpretation.

3. Following the comment of the Referee, we performed completely new calculations for the non-NPF event days, derived average NPF event day-to-non-event day ratios for all variables, discussed and interpreted them in a completely new paragraph in Sect. 4.2, and

– 1 –

added a new Fig. 3 on monthly distribution of the most important variables in 10 panels. Thank to the Referee's comment, these results revealed new aspects of the penomenon. Full exploitation of the overall data set by multistatistical and other methods - including among others the diurnal variations - has been going on in a separate project.

English grammar is good, but sentences are quite long and, frequently do no provide useful information. The length of the text must be shorted (27 pages, without references with only 3 figures and 3 tables).

4. Several sentences were split into parts; the writing and language of the MS was improved, and the MS was shortened. Figures 2 and 4 consist of 4 and 6 panels, respectively, while the supplementary material contains 3 more tables and 2 more figures. We also prepared a new Fig. 3 consisting of 10 panels.

Section 4-2 and Figure 2: It is surprising the monthly evolution of NOx and SO2. Information about DL for SO2 should be provided. Monthly and daily evolution for both NPF and non-NPF events would also provide info about it.

5. The concentration of $SO_2$ in the Budapest area is ordinarily distributed without larger spatial gradients. It suggests that it is usually available for the NPF process. Its monthly average concentrations for NPF and non-NPF days were identical within the uncertainty intervals. The limit of determination (LOD) of the $SO_2$ analyser system applied is approximately 0.2 μg m$^{-3}$. More than 98% of the hourly-mean concentrations were above the LOD. The requested information was added to the text. See also response no. 12.

Section 4.4. As it is do not provide very useful information.

6. Several sentences in this section were removed or shortened.

Methodology: sampling performed at two sites with different inlet configuration; were the particle loses corrected?

7. It was the same experimental system that was deployed at both measurement sites. The inlet tubing (its material, internal and outer diameters, curvature, rain cover, insect net and length) was identical and the DMPS was also the same. It is, therefore, expected that the particle losses in the 2 configurations were very similar. A short note on this was added.

Table 1: add annual mean concentrations in Table 1, and simplify the description in the beginning of section 4.

8.  The concentrations were replaced from the text to Table 1 as requested and the corresponding part of the text could indeed be largely shortened and simplified.

Line 341: define range for UF and N (UF/N)

9.  All size fractions of aerosol particles were defined in lines 171–172 of the MS ver. 2.

Figure captions: can be simplified; too long

10. We simplified or shortened the captions. They should, however, remain self-explanatory, particularly for figures consisting of several panels.

Page 14; lines 439-441: This statement is not directly deduced from the observations.

11. The statement was removed.

Figure 2: scale for NOx and SO2

12. Gas $SO_2$ is a major precursor for NPF and growth events, while there are indications that $NO_x$ can play a suppressing roles in the process. Their concentrations as shown in Fig. 2d for NPF days exhibited largely constant time behaviour. It was indicated by the variability of their monthly medians around the annual median (constant line). Despite this dependency, we would like to keep displaying them as well because atmospheric "observations in this regard are inconclusive" and "there are contrasting observations" (Kerminen et al., 2018). We showed here and are to emphasize by Fig. 2d that in the Budapest area, these gases have limited influence on atmospheric NPF events. The related text was also extended by this information.

Page 26. Lines 774-776: it cannot be inferred from the cited example that NPF can affect urban climate and health risk

13. These lines contain an outlook. They were modified to further emphasize this character.

Page26; rows 792-793: the present research does not permit to identify key chemical players because the methods used e not the most adequate to do it.

14. The sentence was removed.

Imre Salma

[revised manuscript text omitted]

---

## Author Response (AR3)

**Response to Referee number 3**

23 April 2019

The authors thank Referee #3 for his/her expertise and valuable comments to further improve and clarify the MS. We adopted the requested alterations.

Line 447 states that H2SO4 is estimated to 5E05 cm-3 while on line it is claimed that that H2SO4 contributes 12.3% to GR10. Now, 12.3% of a mean GR of 8nm/h is 1 nm/h. To reach a 1 nm/h growth rate at 10 nm size from H2SO4 it needs a concentration of more than 2E07 cm-3 (see e.g. Nieminen et al., Atmos. Chem. Phys., 10, 9773–9779, 2010. Please resolve this contradiction.

A note on the uncertainty of the scaling factor likely as a major source of the difference was added.

Line 706: "The role of H2SO4 in the nucleation process and early particle growth is still determinant or relevant". Where do you show that now? In an earlier version it said "could be"

The sentence was changed back to include the formulation "could be".

**Final message to the Co-Editor, prof. Xavier Querol and**
**to Referees 1, 2, 3, 4 and 6 of the MS**

The authors wish to express their deepest acknowledgement for the devoted, thorough and high-level evaluation work, which helped us to improve and clarify the MS.

Imre Salma